

# Contiguous United States wildland fire emission estimates during 2003-2015

Shawn P. Urbanski[1], Matt C. Reeves[2], Rachel E. Corley[1,3], Robin P. Silverstein[1,3,4], and Wei Min Hao[1]

[1]US Forest Service, Rocky Mountain Research Station, Missoula Fire Sciences Laboratory, Missoula MT 59808, United States

[2]US Forest Service, Rocky Mountain Research Station, Forestry Sciences Laboratory, Missoula MT 59801, United States

10  [3]College of Forestry and Conservation, University of Montana, Missoula MT, 59812, United States

[4]Currently with Montana Department of Public Health and Human Services, Helena, MT, 59620.

*Correspondence to*: Shawn Urbanski (surbanski@fs.fed.us)



**Abstract.** Wildfires are a major source of air pollutants in the United States. Wildfire smoke can trigger severe pollution episodes with substantial impacts on public health. In addition to acute episodes, wildfires can have a marginal effect on air quality at significant distances from the source presenting significant challenges to air regulators' efforts to meet National Ambient Air Quality Standards. Improved emission estimates are needed to quantify the contribution of wildfires to air pollution and thereby
inform decision making activities related to the control and regulation of anthropogenic air pollution sources.

To address the need of air regulators and land managers for improved wildfire emission estimates we developed the Missoula Fire Lab Emission Inventory (MFLEI), a retrospective, daily wildfire emission inventory for the contiguous United States (CONUS). MFLEI was produced using multiple datasets of fire activity and burned area, a newly developed wildland fuels map and an updated emission factor database. Daily burned area is based on a combination of Monitoring Trends in Burn Severity
(MTBS) data, Moderate Resolution Imaging Spectroradiometer (MODIS) burned area and active fire detection products, incident fire perimeters, and a spatial wildfire occurrence database. The fuel type classification map is a merger of a national forest type map, produced by the USDA Forest Service (USFS) Forest Inventory and Analysis (FIA) program and the Geospatial Technology and Applications Center (GTAC), with a shrub and grassland vegetation map developed by the USFS Missoula Forestry Sciences Laboratory. Forest fuel loading is from a fuel classification developed from a large set (> 26,000 sites) of FIA surface fuel
measurements. Herbaceous fuel loading is estimated using site specific parameters with normalized differenced vegetation index from MODIS. Shrub fuel loading is quantified by applying numerous allometric equations linking stand structure and composition to biomass and fuels, with the structure and composition data derived from geospatial data layers of the LANDFIRE Project. MFLEI provides estimates of CONUS daily wildfire burned area, fuel consumption, and pollutant emissions at a 250 m × 250 m resolution for 2003–2015. A spatially aggregated emission product (10 km × 10 km, 1 d) with uncertainty estimates is included to
provide a representation of emission uncertainties at a spatial scale pertinent to air quality modelling. MFLEI will be updated, with recent years, as the MTBS burned area product becomes available. The data associated with this article can be found at https://doi.org/10.2737/RDS-2017-0039.



## 1 Introduction

Annually, open biomass fires are estimated to burn in excess of three million km$^2$ (Giglio et al., 2013) and emit 46.6 Tg of particulate matter (36.6 Tg of fine particulate matter, PM$_{2.5}$) (van der Werf et al., 2017). Globally, the dominant biomass burning regions are sub-Saharan Africa, Brazil, and Equatorial Asia (van der Werf et al., 2017; Wiedinmyer et al., 2011), regions where
fire ignitions are driven by human activity (Andela et al., 2017). In many regions across the globe, biomass fires are a significant source of air pollution and can be a major hazard to public health (Johnston et al., 2012). Fresh biomass smoke is a rich mixture containing hundreds of gases (Hatch et al., 2015; Urbanski, 2014) and particulate matter diverse in size, composition, and morphology (Reid et al., 2005a; Reid et al., 2005b). Fine particulate matter (PM$_{2.5}$) is the smoke constituent presenting the primary public health hazard (Reisen et al., 2015). In addition to PM$_{2.5}$, the photochemical processing of the volatile organic compounds
and nitrogen oxides present in smoke can also produce ozone (O$_3$) (Jaffe and Widger, 2012; Lindaas et al., 2017), another air pollutant which poses a public health threat (Nuvolone et al., 2018). The health impacts associated with exposure to wildfire smoke include increases in respiratory and cardiovascular morbidity and mortality (Fisk and Chan, 2017; Liu et al., 2015; Williamson et al., 2016).

While biomass burning in the contiguous United States (CONUS) is a small contributor to emissions globally, it is a significant
source of air pollution in the US. Wildfire smoke has created severe air pollution episodes with substantial impacts on public health (Fann et al., 2018; Kochi et al., 2012; Rappold et al., 2014). In addition to public health impacts, wildfire smoke presents challenges for air regulators and land managers. Under the US federal Clean Air Act (CAA), the Environmental Protection Agency (EPA) has established National Ambient Air Quality Standards (NAAQS) to protect public health and the environment (USEPA, 2018a). The NAAQS include standards for PM$_{2.5}$ (24 h and annual) and O$_3$ (8 h). The CAA requires states to adopt plans to
achieve NAAQS and control emissions that may impact air quality in downwind states (USEPA, 2013). Thus identifying the contribution of wildfires to air pollution, even marginal impacts at long distances from the fires, is important for air regulatory efforts. For example, Lui et al. (2016) have estimated that on days that exceed regulatory PM$_{2.5}$ levels in the western US, wildfires account for >70% of total PM$_{2.5}$ loading. Ozone production from wildfires impacting both rural and urban areas has been reported. At remote monitoring sites in the intermountain west US, Lu et al. (2016) found that 31% of summertime O$_3$ exceedances (days
when O$_3$ exceeded the 8 h NAAQS) were attributable to wildfires. However, given the complex processes involved in O$_3$ formation, quantifying the amount attributable to fire emissions in urban areas is particularly difficult (Gong et al., 2017; Brey and Fischer, 2016; Jaffe and Wigder, 2012). Air regulators need accurate emission inventories to quantify the contribution of wildfires to air pollution and thereby develop effective and efficient strategies to control anthropogenic air emission sources. Accurate emission inventories also improve the ability of state air regulators to properly identify wildfire induced NAAQS exceedances,
which qualify for treatment under the EPA exceptional events rule (USEPA, 2018b).

Several biomass burning emission inventories that include CONUS are available (van der Werf et al., 2017; Wiedinmyer et al., 2011; French et al., 2014; Zhang et al., 2017). Of these, the global inventories Global Fire Emissions Database (GFED; van der Werf et al., 2017) and Fire INventory from NCAR (FiNN; Wiedinmyer et al., 2011) are probably the most widely used in atmospheric chemistry and air quality modelling. The Wildland Fire Emissions Information System (WFEIS; French et al., 2014)
provides daily fire emission estimates for CONUS for 2001–2013. Given many options, why develop another emission inventory? In terms of wildfire emission estimates for CONUS, we believe the emission inventory presented in this paper, the Missoula Fire Lab Emission Inventory (MFLEI), may improve upon currently available inventories. We are able to employ comprehensive datasets on the distribution and assemblage of vegetation cover and fuel loading (biomass available for combustion) that are available only for CONUS. MFLEI uses a forest type map and a new forest fuel classification, both of which are based on a national
forest inventory dataset, providing more accurate fuel loading estimates compared to the fuels layer used in WFEIS (Keane et al.,



2013). As a retrospective inventory, MFLEI is able to leverage geospatial fire activity information including high spatial resolution burned area and burn severity products that are not available for real-time inventories (e.g. FiNN). Additionally, much of the fire activity data used in MFLEI is produced by US land management agencies and is available only for US territory, and therefore is not used in global inventories. Our inventory is also able to use a large and growing body of published emission factor data to

craft emission factors specifically for fire prone CONUS ecosystems.

Improved CONUS emission estimates will help quantify the contribution of wildfires to air pollution and thereby inform decision making activities related to the control and regulation of anthropogenic air pollution sources. The ability of states to properly identify wildfire induced NAAQS exceedances, which qualify for treatment under the EPA exceptional events rule (USEPA, 2018b), may also be enhanced with an improved inventory. Further, given the benefit of improved fire activity

information, retrospective emission inventories may help identify and diminish deficiencies of real-time emission inventories, which are used to forecast smoke impacts on air quality and reduce risks to public health.

## 2 Methods

### 2.1 Biomass burning emission model

MFLEI provides estimates of daily emissions of $CO_2$, $CO$, $CH_4$, and $PM_{2.5}$ from wildland fires for CONUS. The inventory has a

spatial resolution of 250 m which is established by the MFLEI land cover map (Sect. 2.2). Burned pixels are identified and assigned nominal burn dates using a spatially resolved burned area dataset developed from four fire activity datasets (Sect. 2.3). The land cover classifications of the MFLEI map are used to assign fuel loading (biomass per unit area available for combustion) and combustion completeness to burned pixels. Fuel loading of forested pixels is based on a fuel classification system developed from forest inventory measurements (Sect. 2.4.1). A spatially explicit rangeland fuels map supplies fuel loading for pixels of herbaceous

and shrub cover types (Sect. 2.4.2). The inventory estimates emission intensities for each 250 m grid cell (k) and day (t) using Eq. (1):

$$E_i(\mathrm{k,t}) = \mathrm{EF}(i,k) \times \sum_j F(k,t,j) \times C(k,t,j), \qquad\qquad (1)$$

where $E_i$ is the emission intensity of species i for grid cell k on day t in units of kg-i m$^{-2}$ day$^{-1}$. The driving variables in Eq. 1 are the pre-fire dry fuel loading for fuel component j (F; kg m$^{-2}$), combustion completeness, which is the fraction of fuel component j

consumed by fire on the day the grid cell burned (C; day$^{-1}$), and the emission factor for species i, which is the mass of i emitted per mass dry fuel consumed (EF; kg-i kg$^{-1}$). The inventory assumes each burned grid cell is burned in its entirety on the estimated burn day (Sect. 2.3.2). Fuel loading (F), combustion completeness (C), and emission factors (EF) all depend on grid cell properties. F is assigned based on a grid cell's forest type group or taken from a rangeland fuel loading map in the case of herbaceous and shrub cover types. C depends on fuel type and also on fuel moisture regime and burn severity classification for forest pixels (Sect.

2.6). EF depend on the fuel type (Sect. 2.7). The mass of species i emitted on the day a grid cell burned (EM$_i$; kg-i day$^{-1}$) is the product of the emission intensity ($E_i$) from Eq. 1 and the grid cell area (A), which is 62,500 m$^2$.

### 2.2 Land cover map

The MFLEI land cover map was created by combining a 250 m spatial resolution CONUS forest type group map with a rangelands map. The forest type group map, the USDA Forest Service (USFS) National Forest Type Dataset (Ruefenacht et al., 2008; available

at https://data.fs.usda.gov/geodata/rastergateway/forest_type/), was used as the base map for the MFLEI land cover map. The forest classification accuracy of the USFS forest type group map is generally around 60 to 70 percent (Keane et al., 2013; Ruefenacht et



al., 2008) with a forest/non-forest classification accuracy ranging from 80 to 98 percent (Blackard et al., 2008). Pixels mapped as non-forest in the forest type group map were then assigned a cover type of shrub, herbaceous, or non-fuel using the CONUS rangelands product of Reeves and Mitchell (2011). The MFLEI cover type map is shown in Fig. 1 and the cover type descriptions are provided in Table 1. During burned area mapping (Sect. 2.3.1) the land cover type codes of the MFLEI are used to assign the
fuel codes listed in Table 1 to burned pixels. Three of the mapped cover types were forest type groups for which there was insufficient data to develop a fuel loading classification (Sect. 2.4.1). Therefore, during the burned area processing, the fuel codes associated with these cover types, 1380, 1980, and 1990, were recoded as 1360, 1950, and 1950, respectively. Also during processing of the burned area data, the fuel codes of forest pixels in the eastern US that were classified as 1180, 1700, 1900, and 1950 were recoded to 2180, 2700, 2900, and 2950, respectively. This was done because the forest inventory surface fuels dataset
used to develop fuel classifications (Sect. 2.4.1) indicated substantially different fuel loadings between eastern and western (11 western states) forests for these forest type groups. Burned grid cells classified as non-fuel in the land cover map were assigned a fuel load = 0 and did not produce emissions. In post-emission processing of the dataset, the non-fuel, zero emission burned pixels were assigned a cover type classification from the National Land Cover Database 2011 (NLCD) (Homer et al., 2015). This was done to track wildfire impacts on agricultural and developed lands or identify possible agricultural burning. Pixels that were not
classified as forest or rangeland in the MFLEI land cover map were fixed as 'No Data' when the NLCD dataset classification was forest, herb, or shrub.

The focus of MFLEI is wildfires, which are fires resulting from unplanned ignitions (e.g. lightning, arson, accidents). The other types of open biomass burning common in CONUS are prescribed fires and agricultural fires. We define agricultural fires as the burning of crop residue or preparation of fields for planting. Croplands are classified as non-fuel in the MFLEI land cover map
and are assigned zero emissions in the inventory. Prescribed fires are intentionally ignited to achieve land management objectives (e.g. hazardous fuel reduction, ecosystem restoration, and preparation of rangeland for grazing). Prescribed fires are not excluded from MFLEI, although given the focus on wildfires they are certainly underrepresented as discussed in Section 3.5.

## 2.3 Burned area

Burned area was derived from MODIS and Landsat based burned area products, a dataset of fire perimeter polygons mapped to
support fire management activities, and a fire occurrence database. Burn dates were primarily assigned based on the MODIS burned area product and active fire detection products from MODIS and the Visible Infrared Imaging Radiometer Suite (VIIRS). When a burn date could not be assigned from MODIS or VIIRS data, it was estimated from generalized fire activity cycles and the fire size and duration obtained from the fire occurrence database or other administrative records.

### 2.3.1 Burned area mapping

On an annual basis, potentially burned grid cells of the MFLEI land cover map were identified by an overlay of burned area polygons and rasters in ArcMap. Four burned area/fire activity datasets were used to extract potentially burned pixels: Monitoring Trends in Burn Severity (MTBS) fire boundaries (MTBS, 2017a; Eidenshink et al., 2007), the MODIS active fire based Direct Broadcast Monthly Burned Area Product MCD64A1 (MCD64) (MCD64A1, 2016; Giglio et al., 2009), incident fire perimeters from the Geospatial Multi-Agency Coordination Wildland Fire Support archive (GEOMAC, 2015), and a spatial wildfire
occurrence database (FOD) (Short, 2017).

The MTBS project maps fire boundaries and burn severity for large fires (> 404 ha in the west and > 202 ha in the east) across the US from 1984 to the present (Eidenshink et al., 2007; MTBS, 2017c). MTBS fire boundaries are polygons representing burned area detected from post-fire Landsat TM/ETM/OLI imagery (Eidenshink et al., 2007). The polygon attributes for each MTBS





boundary include a unique fire ID, fire start date, and fire name. The MTBS fire ID attribute was used to aggregate burned grid cells by fire event and to filter the FOD point dataset to avoid double counting of fires. The primary MTBS product is thematic burn severity rasters, which classify burn severity within the fire boundaries (Eidenshink et al., 2007; MTBS, 2017b). We used the MTBS burn severity rasters to identify unburned regions within MTBS fire boundaries and to develop scaling factors to

approximate unburned patches for burned area mapped using MCD64, GEOMAC, and FOD, as described in Sect. 2.3.3.

The MCD64 product maps burned areas using 500 m MODIS imagery coupled with 1 km MODIS active fire detections (Giglio et al., 2009). MCD64 is a monthly, 500 m resolution raster product that provides an estimated burn date for each pixel identified as burned. We used MODIS Collection 5.1 of MCD64A1 (MCD64A1, 2016). The most recent version of the MCD64A1 product, Collection 6, became available in January 2017 (Giglio et al., 2015). The MCD64 product is the primary burned area data source

for the Global Fire Emission Database (GFED) (Giglio et al., 2013) during the MODIS era. Details for accessing the product can be found on the GFED website: http://www.globalfiredata.org/ (last access: June 4, 2018).

The GEOMAC dataset is a collection of fire perimeter polygons. For large fire events, fire perimeters are periodically mapped by incident management teams, typically using airborne infrared imagery. These incident perimeter polygons are produced to support fire management activities. Since their purpose is identifying the fire perimeter, not mapping the actual area burned, the

area within a perimeter typically includes unburned regions. We attempt to compensate for this as discussed in Sect. 2.3.3. For these reasons, we give the MTBS dataset precedence over the GEOMAC. Further discussion regarding the use of incident perimeters as 'ground-truth' burned area may be found in Urbanski et al. (2009) and Key and Benson (2006). Final fire perimeters from the GEOMAC dataset were checked against the MTBS fire boundaries using the products' fire name attributes to remove GEOMAC perimeters for fires present in the MTBS dataset.

FOD is a spatial database of wildfires that occurred in the United States from 1992–2015 generated from wildfire records acquired from the reporting systems of federal, state, and local fire organizations (Short, 2017). FOD provides a point location for each fire, not a spatial object that maps burned area. Other FOD dataset attributes used in our analysis include final fire area, discovery date, containment date, fire name, fire code, and the MTBS Fire ID attribute from the MTBS perimeter dataset (MTBS fires only). We used the FOD dataset to capture fires not included in the MTBS, GEOMAC, or MCD64 datasets. We filtered the

FOD dataset for fires contained in either the MTBS or GEOMAC datasets using the MTBS Fire ID or the fire name and fire code attributes (for GEOMAC) from the datasets. Fires < 4 ha in size were also removed due to their minor contribution to total burned area; while fires < 4 ha accounted for 86 % of all fires in the FOD database for 2003–2015, they only comprised 1.5 % of total fire area. Finally, FOD fires with locations that fell within a distance $D_f$ ($D_f = 2\sqrt{A/\pi}$ , where A is the FOD fire area) of any grid cell identified as burned by either the MCD64 or GEOMAC datasets were removed. Following these filtering actions, MFLEI land

cover map grid cells within a distance $D_f/2$ of an FOD fire location were flagged as burned.

### 2.3.2 Burn date assignment

Of the four datasets used to map burned area, only MCD64 provides an estimated burn date, and these were assigned to MFLEI grid cells identified as burned by the MCD64 product. Grid cells identified as burned by the MTBS, GEOMAC, or FOD datasets were assigned an estimated burn date as follows. First, all grid cells (non-MCD64 sourced) were assigned a fire start date and,

when available a fire containment date, on a fire event basis. The MTBS, GEOMAC, and FOD datasets include fire event identifiers and fire start dates (or discovery dates) which were added as attributes to burned grid cells. The FOD dataset also includes a containment date for many fire events and it was added as an attribute to burned MFLEI grid cells when available. Most of the fires in the MTBS and GEOMAC dataset are also included in FOD. Fire event identifiers, MTBS Fire ID, and the fire name and fire code attributes from GEOMAC, were used to associate MTBS and GEOMAC sourced burned pixels with FOD fire events and



thereby assign containment dates when available. Next, grid cells identified as burned by the MTBS, GEOMAC, or FOD datasets were assigned an estimated burn date using one of the following methods in order of precedence:

1) Grid cells within 500 m of a MCD64 sourced pixel were assigned that pixel's burn date.

2) Grid cell burn dates were assigned from MODIS active fire detections (MCD14) (Giglio et al., 2003) using spatial and temporal proximity criteria to associate active fire detections with burned grid cells. We assigned each active fire detection a spatial buffer, $X_b$, which defines the maximum distance at which it can be associated with a MFLEI grid cell for purposes of ascribing a burn date. MCD14 pixels have nominal dimensions of 1 km × 1 km; however, the actual size and location of a detected active fire is unknown. In consideration of this spatial uncertainty, we assigned $X_b$ a default value of 2 km. The dimensions of MCD14 pixels are 1 km × 1 km at nadir, but increase with distance off nadir, reaching 4.8 km (scan direction) × 2 km (track direction) on the edges of the MODIS scanning swath (Nishihama et al., 1997). For off nadir pixels, $X_b$ was set to the dimension of the scan direction when > 2 km (pixel dimensions were among the attributes of the MCD14 product used in analysis). For each burned grid cell, we identified the nearest active fire detection located within a distance $X_b$ and falling in the time frame: ($D_{start}$ − 3 days) to ($D_{cont}$ + 3 days), where $D_{start}$ and $D_{cont}$ are the grid cell's fire start date and fire containment date attributes. The temporal criteria was used to eliminate any active fire detections from an unrelated fire that occurred during a different time period. For the years 2014 and 2015, VIIRS I-band active fire detections (Schroeder et al., 2014) were also used to assign pixel burn dates. The procedure was similar to that used with the MCD14 product, except that the VIIRS active fire detection spatial buffer, $X_b$, was set to 750 m, which is twice the spatial resolution (375 m) of VIIRS I-band pixels at nadir. Because the VIIRS I-band active fire detection product has significantly superior mapping capabilities compared to the MCD14 product (Schroeder et al., 2014), it was given precedence over MCD14 for assigning pixel burn dates. Burned grid cells not associated with MCD64 were assigned a burn date equal to the date of the nearest active fire detection meeting the above spatial and temporal criteria. The MCD14 and VIIRS I-band active fire data used was obtained from the USDA Forest Service Remote Sensing Application Center's Active Fire Mapping Program (https://fsapps.nwcg.gov/afm/gisdata.php).

3) Event based extrapolation. Following burn date assignment steps 1 and 2, 28% of the burned grid cells were without burn dates. Forty-six percent of these undated grid cells were associated with fire events which had some grid cells that did have burn dates. For these fire events, grid cells without burn dates were assigned the burn date of the nearest grid cell with a burn date.

4) The final step for assigning burn dates addressed burned grid cells of "dateless" fire events, those without any burn date associated with the grid cells. In order to assign estimated burn dates to these grid cells, which comprised 15% of all the grid cells, we developed what we refer to as "burn day distributions". These are empirical distributions of the fraction of event total burned area as a function of days since ignition. One set of burn day distributions was derived using MTBS fire events which had a containment date and also had > 95% of grid cells assigned a burn date in steps 1 or 2 above. From these fire events, burn day distributions were created according to six fire size classes (in ha): 200–625, 625–1250, 1250–3125, 3125–6250, 6250–12,500, 12,500–25,000. The burn day distribution for the 12,500–25,000 ha size class is shown in Fig. A1 and the distributions for all six size classes are provided in the dataset supplement (file\Supplements\BurnDayDist.csv, see Sect. 4). The burned grid cells of dateless fire events > 200 ha in size were assigned burn dates using the burn day distribution for the appropriate size class. For fire events with a containment date, the burn day distribution was truncated to correspond to the fire



duration (containment date - fire start date) and normalized. When a dateless fire event was < 200 ha and had a containment date, grid cell burn dates were assigned one at time cycling through the days between the fire start date and the containment date in chronological order until all grid cells were assigned. Fire events < 200 ha and without containment dates were assigned durations using Table A1 and the burned grid cells were distributed one per burn day by cycling through the burn days in chronological order until all grid cells were assigned.

### 2.3.3 Unburned and lightly burned grid cells

Wildfires typically do not impact fuels uniformly across the landscape and it is not unusual for significant area within a fire perimeter to be unburned or only lightly burned (Kolden et al., 2012). MTBS burn severity thematic classifications were used to account for unburned or lightly burned regions (MTBS, 2017b). The MTBS burn severity thematic classifications were developed to represent fire effects on above-ground biomass (Eidenshink et al., 2007; Schwind , 2008). MTBS assigns six burn severity classifications (BSEV) to pixels within fire boundaries: 1) unburned to low burn severity, 2) low burn severity, 3) moderate burn severity, 4) high burn severity, 5) increased green, 6) no data. We elected to designate BSEV = 1 as unburned, which is consistent with MTBS program publications that describe this classification as areas which are either unburned or where visible fire effects occupy < 5 % of the site at the time of observation (Schwind, 2008). MFLEI burned grid cells associated with a fire analyzed by the MTBS project were compared against a coarse scale MTBS thematic burn severity map (30 m original resampled to the MFLEI 250 m grid using majority sampling). Coarse scale MTBS pixels classified as BSEV = 5 or BSEV = 6, increased green or no data, respectively, were randomly re-assigned a value between 1 and 4. This reassignment was conducted on a fire event basis in proportion to the frequency of pixels originally classified BSEV 1–4. MFLEI grid cells classified as BSEV = 1, "unburned to low severity", in the coarse scale MTBS product were flagged as unburned. MFLEI burned grid cells not associated with a fire analyzed by the MTBS project were randomly assigned a BSEV value based on a generic cover type–BSEV empirical distribution developed from the CONUS wide MTBS thematic classification maps for 2003–2013. The cover type–BSEV distribution is shown in Table 2.

### 2.4 Fuel loading

Fuel loading was represented with the 14 fuel components in Table 3. Models of forest fuel loading were developed using data from the USFS Forest Inventory and Analysis (FIA) National Program as described in Sect. 2.4.1. The rangeland fuel product (Sect. 2.4.2) provided spatially explicit fuel loadings for grassland and shrub ecosystems.

### 2.4.1 Forest fuel loading

**Surface fuel loadings**

We developed an expanded version of the Fuels Type Group (FTG) fuel classification system assembled by Keane et al. (2013) using recently available FIA fuels data and also including plot data from the eastern US. The FIA inventory is comprised of three phases of data collection as described in Bechtold and Patterson (2005). The inventory is designed to cover forested land (10 % stocked with tree species, see Bechtold and Patterson (2005)), of all ownership across the US. Phase 1 sampling provides information to stratify inventory ground plots and improve the precision of estimates of population totals (Bechtold and Patterson, 2005). In Phase 2, measurements are taken on the standard FIA base grid which has a density of approximately 1 sample location per ~ 2428 ha (6000 acres). Phase 2 collects information such as height and diameter of standing trees and physiographic class and land ownership. Phase 3 involves sampling of forest health indicators, such as the down woody material (DWM) indicator. The DWN indicator estimates dead organic materials including downed woody debris, litter, and duff (Woodall and Monleon,





2008). The DWM indicator was used to estimate plot level surface fuel loading as described below. Phase 3 sampling is conducted on a subset of Phase 2 plots (approximately 1/16 of Phase 2 plots). In the western US, the FIA units began collecting the DWM indicator on all of their Phase 2 plots in the early 2000's (Keane et al., 2013), thus the density of surface fuel plots used to assemble the FTG classification is significantly higher in the west. Fig. 2 maps the locations of the FIA plots used to develop the expanded
FTG surface fuel classification for MFLEI.

Our FTG classification is based on 27,124 plots compared with 13,138 used in Keane et al. (2013). We used only single condition plots, plots where all four subplots were the same condition (land class, reserved status, owner group, forest type, stand-size class, regeneration status, and stand density) (O'Connell et al., 2016). The FTG classification summarizes fuelbed component loadings (Table 3) by FIA forest type groups using fuels data from the FIA Database acquired from the FIA DataMart website
(https://www.fia.fs.fed.us/tools-data; FIA, 2015). Five tables were accessed from the FIA dataset: REF_FOREST_TYPE, COND, PLOT, COND_DWM_CALC, and DWM_COARSE_WOODY_DEBRIS. A detailed description of these tables is provided by O'Connell et al. (2016). For an in-depth description of the FIA sampling design, estimation, and analysis procedures see Woodall and Monleon, 2008, O'Connell et al., 2016, and Woodall et al., 2013, and for an abbreviated summary see Keane et al. (2013). Data assembled from the COND_DWM_CALC table included loading (biomass per unit area) of fine woody debris by three size
classes: small, medium, and large (Table 3), duff loading and depth, and litter loading and depth. Data from the DWM_COARSE_WOODY_DEBRIS table was assembled to provide loadings of coarse woody debris by eight size/decay class combinations (Table 3) following the methods described in Woodall and Monleon (2008). Best estimate loadings of the surface fuel components were taken as the average values of all plots for each fuel classification and are shown in Table 4. The surface fuel loading data for the 27,124 plots used to develop Table 4 and to derive uncertainty estimates in the emission modeling (Section
2.9) are included in the MFLEI dataset (file \Supplements\Fuel_Load_Plot_Data.csv, see Sect. 4). The MFLEI land cover type map assigns an FTG to all forest pixels. Four FTG, 180, 700, 900, and 950, had significant fuel loading differences between western (11 western states) and eastern plots. Therefore, separate fuel classifications, west and east, were made for these FTG and they are differentiated by the fuel code (Table 1) which is assigned during burned area mapping as described in Sect. 2.2. As discussed in Keane et al. (2013), the variability of surface fuel loading within FTGs is quite large. Figure 3 plots the distribution
of surface fuel loading for the FIA plots of three FTG, Loblolly/shortleaf pine (160), Douglas-fir (200), and California mixed conifer (370). The surface fuel loading plot data have a log-normal like distribution with long tails. The high variability in surface fuel loading is the primary source of uncertainty in the emission estimates for forest fires (Section 2.9).

**Understory fuels**
The loading of forest understory fuels, shrubs (vascular plants with woody stems that are not defined as trees by FIA Phase 2) and herbs (non-woody vascular plants including but not limited to ferns, moss, lichens, sedges, and grasses), was derived from raster maps of forest understory carbon (Wilson et al., 2013). The raster maps of forest understory carbon were combined with the USFS FIA Forest Type Group map (Ruefenacht et al., 2008) to derive empirical distributions of understory fuel loading for each FTG class (assuming a biomass carbon content of 50%). The fuel loading distributions were used to provide uncertainty estimates for
the emission modeling (Sect. 2.9). Partitioning of the understory fuel loading between shrubs and herbs was based on herb to shrub ratios from the Fuel Characteristics Classification System (FCCS) and First Order Fire Effects Model (FOFEM) reference fuel models (Ottmar et al., 2007; Riccardi et al., 2007; Lutes, 2016a). The empirical distributions of understory fuel loading for all FTG classes are included in the MFLEI dataset (file \Supplements\Understory_Fuel_Dist.csv, see Sect. 4). Best estimate loadings for herb and shrub fuel components were taken as the average values of all plots for each fuel classification and are shown
in Table 4.



**Canopy fuels**

Available canopy fuel (ACF), the dry mass of canopy fuels likely to be consumed in a fully active crown fire (needles, lichen, moss, and live and dead branch wood ≤ 6 mm in diameter) (Scott and Reinhardt, 2001), was derived from FIA plot Treelist tables.

FIA Treelist tables (which are named TREE in the FIA database) provide a detailed inventory of trees on FIA plots (O'Connell et al., 2017). FIA plots with Treelists are based on Phase 2 sampling which are far more numerous than the Phase 3 plots used to derive surface fuel loadings (see above). We used the Treelist table variables: species code (SPCD), diameter (D), crown class code (CCLCD), tree status (STATUSCD), and tree density (TPA) to estimate ACF associated with each Treelist table entry using empirical equations from the literature following the approach outlined in the FuelCalc User's Guide (Lutes, 2016b). FuelCalc is

a fuel management software system which can be used to calculate forest canopy characteristics at an inventory plot. For each of 363,060 FIA plots with a Treelist, stand level ACF was calculated using Eq. 2:

$$ACF_{stand} = \sum_{i}^{N}(acf_i \times TPA_i), \qquad (2)$$

where the subscript i is the index for the softwood tree species in the stand and $acf_i$ and $TPA_i$ are tree level available canopy fuel and tree density. $acf_i$ and $TPA_i$ are calculated as described in the Supplement. $ACF_{stand}$ were aggregated by FTG (an FIA plot

variable) and the mean was taken as the best estimate which are listed in Table 4. The $ACF_{stand}$ aggregated by FTG were fit to Weibull probability distribution functions to derive uncertainty estimates for the emissions modeling (Sect. 2.9). Best estimate ACF and optimized parameters for fits to Weibull probability distribution functions (PDF) are provided in Table B1.

**Total forest fuel loading**

Average forest fuel loading is dominated by the surface fuels for all forest fuel types (25 FTG plus 4 eastern variants (see Sect. 2.2)), as shown in Figure 4. Greater than 70% of total fuel loading resides in the surface fuels for 25 of the 29 forest fuel types. Surface fuel components (Table 3) are often grouped into litter, fine woody debris (fwd; down dead wood with diameter < 7.62 cm), coarse woody debris (cwd; down dead wood with diameter >=7.62 cm), and duff. These groupings reflect the surface to volume ratio of the fuel particles, an important determinant in the rate of fire spread (Scott and Burgan, 2005) as well as the

combustion characteristic of the fuels. Litter and fine woody debris tend to favor flaming combustion while coarse woody debris, and duff especially, favor smoldering combustion processes (Urbanski, 2014). Figure 5 plots the fraction of total fuel load residing in duff, litter, fine woody debris, and coarse woody debris for the 29 forest fuel types.

**2.4.2 Rangeland fuel loading**

Rangeland fuels were estimated using the Rangeland Vegetation Simulator (RVS) (Reeves, 2016) and began with delineating the

spatial domain of rangelands in CONUS (land cover type codes 1 and 2 in Figure 1), as described in Reeves and Mitchell (2011), and constrained using the forest type map developed by Blackard et al. (2008). If a forest type was indicated for a given pixel in the Blackard et al. (2008) map, no rangeland fuel data were estimated for that pixel. The vegetation form (herbaceous or shrub) and type (e.g. Chihuahuan Mixed Desert and Thornscrub) were assigned from the Landfire Project (LF) Existing Vegetation Type (EVT) geospatial data layer (LANDFIRE, 2016). Different methods were used to quantify woody and herbaceous fuels (Figure

35 6).





**Shrub**

The derivation of shrub fuel loading used two LF products in addition to EVT as input: Existing Vegetation Height (EVH) and Existing Vegetation Cover (EVC). The height estimates at each pixel in the EVH product are thematic classes representing a range of potential heights (Table C1). The range of potential heights provided by the EVH enables three values of shrub fuels to be

estimated at each pixel (median, maximum, and minimum). EVC represents the vertically projected percent cover of the live canopy.

Generation of shrub fuel loading data involves several steps (Fig. 6) which are briefly described here. Details of the approach are illustrated in Appendix C. First, crown dimensions are derived from EVH and the projected crown area on a horizontal surface (PCH), the latter of which is estimated using Eq. 3 (Frandsen, 1983):

$$log_{10}(\text{PCH}) = -0.8471 + 2.2953 log_{10}(HT), \quad\quad\quad (3)$$

where PCH is in cm$^2$ and HT is the estimated height class of shrubs in cm at each pixel (from the EVH product). Crown dimensions are then used in one of 31 species specific equations from the RVS allometric library to estimate per stem biomass (PSB; kg stem$^{-1}$).

Next, the estimate of stem density (SD) at each pixel, (stem ha$^{-1}$) is used to expand PSB to a per-area basis. SD is estimated as:

$$\text{SD} = (\text{PCH} / 10^8) * \text{EVC} \quad\quad\quad (4)$$

where SD is stem density, and the value $10^8$ converts cm$^2$ to a per hectare basis. In effect, the number of times PCH can be divided into a hectare is scaled by the canopy cover (EVC). The total shrub biomass (TSB; kg ha$^{-1}$) is the product of PSB and SD. This four step process was conducted at each pixel using the minimum, maximum, and median shrub heights from EVH (Table C1) to provide lower, upper, and middle estimates of fuel loading, respectively.

**Herb**

The derivation of herb fuel loading used the EVT and MODIS growing season maximum Normalized Difference Vegetation Index (NDVI), and the Soil Survey Geographic (SSURGO) annual productivity map, which consists of polygons with estimates of rangeland productivity (dry-weight/area/year) for normal, favourable, and unfavourable production years (Soil Survey Staff, 2015). The SSURGO productivity data were derived from the USDA National Resource Conservation Service soil survey geographic

database (Soil Survey Staff, 2015). Herbaceous biomass is estimated as a function of the annual maximum NDVI across 51 grassland vegetation types. The three SSURGO production values reported at each soil polygon were paired with the average, minimum, and maximum NDVI values (from 2000–2016) for each of the 51 vegetation types dominated by herbaceous species (Fig. 7). When this relationship is applied for each year in the time series between 2000 and 2016, an annual estimate of rangeland production can be made at every pixel. The present year's herbaceous production (from 2000–2016) is added to estimated standing

dead herbaceous vegetation ("holdover") resulting from previous growth (see below). Annual production added to the holdover from previous years creates the 'herbaceous loading' pool (HL; Fig. 6).

Estimating the previous year's standing dead or herbaceous litter material is based upon experimental (Irisarri et al. 2016) and anecdotal observations. This topic is not widely studied across multiple ecosystems and it is difficult and time consuming to derive experiments that track the fate of herbaceous growth, senescence and decomposition across multiple vegetation types. The paucity

of suitable plot data for estimating the amount of standing dead material is therefore based on observations of various vegetation



stands with significant herbaceous components throughout the western US. In addition, the USDA Agricultural Research Service (ARS) recently provided results from 10 years of grassland observations on shortgrass steppe near Cheyenne, Wyoming and standing dead values averaged 22% across treatments. This means that, on average, in shortgrass steppe, standing crop of the present year includes 22% of the previous year's production plus the present annual production. The function used in the RVS to

estimate the standing dead material is $y = 100e^{-1.495x}$, which yields values of 22% at year 1 and 5% at year 2.

To capture the range of variability of the herbaceous response, the coefficient of variation (C.V. = mean / standard deviation of the annual production between 2000 and 2016) was applied at each pixel dominated by herbaceous lifeforms. This yields three potential values of herbaceous loading at each pixel (mean, mean +/- C.V.). Likewise the range of standing dead values over 2000–2016 was estimated using the mean +/- C.V. At this stage herbaceous loading (HL) and shrub loading (TSB) have been

produced and are mosaicked together to form a seamless depiction of fuels and are available for simulation of fuel consumption and emissions. Raster files of the herbaceous C.V. and the shrub minimum and maximum are included in the MFLEI dataset.

### 2.4.3 Total fuel loading

Best estimate total fuel loading of both forests and rangelands are mapped in Figure 8. Forest fuel loadings range from 1.3 to 13.3 kg m$^{-2}$ (Table 4, Fig. 4). Fuel loadings are considerably less for rangelands, varying from ~0.1 to 5.2 kg m$^{-2}$, with a median value

of 1.8 kg m$^{-2}$. Regions without a mapped fuel loading are classified as non-fuel and are largely agriculture, barren, developed lands or water.

### 2.5 Fuel conditions

Fuel moisture content is a key driver of fuel consumption, especially for coarse woody debris and duff. The National Fire Danger Rating System (NFDRS; Cohen and Deeming, 1985) provides fuel moisture models that classify dead fuels by time lag intervals

which are proportional to the fuel particle diameter. The NFDRS classifications for dead fuel moisture are 1 h, 10 h, 100 h, and 1000 h corresponding to diameters of < 0.64, 0.64–2.54, 2.54–7.62, > 7.62 cm. The algorithms used to simulate surface fuel consumption require fuel moisture content for 1000 h time lag fuels and duff. Surface fuel consumption was simulated for the four fuel moisture regimes shown in Table 5. In the emission modeling, MFLEI grid cells were assigned the 1000 h time lag or 100 h time lag fuel moisture content of the nearest NFDRS station for day of concern. 1000 h fuel moisture content is considered a proxy

for coarse woody debris (see Table 3). Data for NFDRS stations was obtained from the USFS Wildland Fire Assessment System (WFAS) (Wildland Fire Assessment System, 2015) data archive. Missing values were filled by linear interpolation across days. Duff moisture content was estimated using the 100 h fuel moisture content and empirical relationships of Harrington (1982).

### 2.6 Fuel consumption

Best estimates and ranges of consumption (i.e. combustion completeness), for forest surface and understory fuels for the four

moisture regimes used in the emission inventory are shown in Table 6. The best estimate values are based on simulations using algorithms from the fire effects models CONSUME (Prichard et al., 2006) and FOFEM (Lutes, 2016a). The ranges, which were used to estimate uncertainty in the fuel consumption simulations, were assigned as 10-20%. The best estimate and range for the fraction of forest canopy fuel consumed was based on each pixels' burn severity classification (Table 7), which were assigned as described in Sect. 2.3.3. Fuel consumption for shrub and herbaceous grid cells used the natural fuel equations from CONSUME

(Prichard et al., 2006). The rangeland fuel consumption equations used do not include fuel moisture content and therefore were independent of the moisture regime.



### 2.7 Emission factors

The composition and intensity of emissions produced by biomass burning varies with the relative mix of flaming and smoldering combustion. Modified combustion efficiency (MCE), the molar ratio of emitted $CO_2$ to the sum of emitted $CO_2$ and CO (MCE $= \Delta CO_2/(\Delta CO_2 + \Delta CO)$), is a widely used measure of the relative mix of flaming and smoldering combustion. Because the emission

factors (EF) of many species are correlated with MCE, it is a useful metric for extrapolating emissions factors from one set of combustion conditions to another (Urbanski, 2014; Akagi et al., 2011). The MCE observed for wildland fires varies significantly across fire types, for example average MCE values are around 0.94 and 0.93 for rangeland and southeastern forest fires, respectively, but ~0.88 for wildfires in western forests (Urbanski, 2014). This difference in fire properties was accounted for in the emission inventory by using three sets of EF (southern forests, western and northern forests, and rangelands). Data from several

field studies (Table S4) was used to model EF as a linear function of MCE for forest and rangeland fires (Table 8). The linear functions were combined with best estimate MCE values to derive the EF used in the inventory (Table 9). Since the focus of MFLEI is wildfires, the best estimate MCE used for western and northern forests is based on western wildfires. Sufficient field measurement data of MCE and EF for southern wildfires could not be found in the literature. Therefore, the EF used for southern forest fires are based on the large body of prescribed fire studies in the literature. The linear functions and their standard errors in

Table 8 were combined with MCE values, sampled from a normal distribution to account for within fuel group uncertainty (Table 9), to provide an estimate of the uncertainty in the EF which was used in the emission modeling uncertainty analysis (Sect. 2.9).

### 2.8 Emission estimates

The best estimates of fuel loading for the 14 fuel components ($F_{k,j}$, Table 3) were assigned to forest pixels using the mapped forest type group and associated fuel code (Sect. 2.2) and the FTG fuel classification system (Table 4). The fuel code, fuel moisture

regime, and burn severity classification were used to designate combustion completeness by fuel component for each pixel ($C_{k,j}$) using the best estimates from Table 6 and Table 7. Fuel loading for herbaceous and shrub pixels ($F_{k,j}$) was taken from the rangeland fuels map (Sect. 2.4.2). Herbs and shrubs were treated as single component fuels with a combustion completeness that is independent of fuel moisture regime and burn severity classification. $EF_{k,i}$, were selected from Table 9 based on the fuel type and then the best estimate emission intensities for $CO_2$, CO, $CH_4$, and $PM_{2.5}$ were calculated using Eq. 1.

### 2.9 Uncertainty estimates

A Monte Carlo style analysis following the general approach outlined in the IPCC Guidelines for National Greenhouse Gas Inventories (Eggleston et al., 2006) was used to estimate the uncertainty in emission intensities (kg m$^{-2}$) at the pixel level. The method involved randomly selecting a sample of N input values ($\mathbf{X_i}$, $\mathbf{X_{i+1}}$,…,$\mathbf{X_N}$) for the emission model (Eq. 1) and calculating emission intensities ($\mathbf{E_i}$, $\mathbf{E_{i+1}}$,…,$\mathbf{E_N}$), where $\mathbf{X_i}$ is the array of input values needed for a single emission calculation: fuel loading by

component (Table 3), combustion completeness (Tables 6 & 7), and EF ($EFCO_2$, $EFCO$, $EFCH_4$, $EFPM_{2.5}$), and $\mathbf{E_i}$ is the array of emission intensities for $CO_2$, CO, $CH_4$, and $PM_{2.5}$. The samples of input variables were generated based on each pixel's fuel code using the methods summarized in Table 10 and described in more detail below. The value of N was 500 for rangeland pixels. For forest pixels, N was taken as the greatest of 500 or $N_{plots}$, where $N_{plots}$ is the number of plots in the FIA dataset for a given pixel's forest fuel code (Table 4). Next, quantiles (q = .05, .10, .25, .50, .75, .90, .95) of the emissions ($E_{q,b}$) were calculated and saved.

The process was repeated B times, yielding $E_{q,1}$,…$E_{q,B}$ and mean values ($\sum_1^B E_q/B$) were calculated to provide uncertainty estimates of the emissions. Convergence of the distributions was achieved with B = 2000.

Forest surface fuels were generated by using fuel loading arrays sampled from the FIA plot data (included in the MFLEI dataset: file\Supplements\Fuel_Load_Plot_Data.csv, see Sect. 4), i.e. each element i used surface fuel components from a single FIA plot.



This approach was chosen to preserve any correlations among surface fuel components. Uncertainty in the assigned moisture regime and burn severity classification, which are used to determine surface and canopy fuel consumption, respectively, were not considered in this analysis. Therefore, uncertainty analysis produced 464 sets of quantiles for forest pixels (29 forest fuel codes, four moisture regimes, and four burn severity classifications). Burned forest pixels were assigned sets of quantiles based on forest

fuel code, moisture regime and burn severity classification.

The variability in pixel level shrub fuel loading was simulated using means and standard deviations based on the maps of the mean, minimum, and maximum loading (Sect. 2.4.2); with the standard deviation in loading estimated as half the range in maximum and minimum loading at each pixel. To reduce computational demands, shrub pixels were aggregated into bins of mean loading in 50 g m$^{-2}$ increments (50 to 5500 g m$^{-2}$). For each 50 g m$^{-2}$ increment in mean loading, simulations were conducted using

25 increments of standard deviation each corresponding to 10 percentage points of the mean loading value (10% to 250%), resulting in 2750 fuel loading elements (pairs of μ and σ). Similarly, pixel level variability in herbaceous fuel loading was simulated based on pixel specific mean and standard deviation from the maps of the mean and the coefficient of variation (C.V. = σ/μ) of loading (Sect. 2.4.2). As with the shrub fuel loading, the herbaceous pixels were aggregated to reduce computational demands. Herbaceous pixels were grouped according to mean loading by 25 g m$^{-2}$ increments (25 to 500 g m$^{-2}$). For each 25 g m$^{-2}$ increment in mean

loading, simulations were conducted using 22 increments of standard deviation corresponding 5 percentage points of the mean loading value from 5% to 110%, providing 440 fuel loading elements (pairs of μ and σ). Using the general approach described in the first paragraph of this section, a set of emission quantiles were produced for each of the 2750 shrub and 440 herbaceous fuel elements, with fuel loading simulated using a truncated normal distribution with the element's μ and σ, and the combustion completeness and EF using probability distributions described in Table 10. Since rangeland fuel consumption was estimated

independent of moisture regime and burn severity classification, these variables were not considered in the uncertainty analysis. Each burned rangeland pixel was assigned a set of emission quantiles from the simulations based on its cover type (herb or shrub), fuel loading, and fuel load uncertainty.

The spatial and temporal resolutions required of fire emission inventory systems depend on the specific applications for which they are being used. For regulatory related air quality modeling the EPA recommends a horizontal grid resolution of ≤ 12 km for

O$_3$ and PM$_{2.5}$ NAAQS (USEPA, 2007). Therefore the uncertainty estimation approach described above was applied aggregating burned pixels to a 10 km × 10 km grid. The resultant dataset at 1 d and 10 km spatial resolution provides a more relevant representation of the uncertainties of the emissions when used in typical air quality applications. The approach followed that outlined above, except that the emission intensities for each sample, $E_i$, were the sum of emission intensities for all pixels within each 10 km × 10 km grid cell on a given day. Only grid cell days with > 4 burned pixels were considered for this uncertainty

analysis, this excluded 9% of burned area over the 2003–2015 period.

## 3. Results

### 3.1 Annual, seasonal, and monthly

The MFLEI annual burned area, fuel consumed, and PM$_{2.5}$ emitted for 2003–2015 are shown in Figure 9. The average area burned was 22,891 km$^2$ y$^{-1}$; forest accounted for 44% of burned area with the balance split between herb (29%) and shrub (27%) cover

types. The maximum annual burned was 40,714 km$^2$ in 2011 which was >5 times the minimum of 7688 km$^2$ in 2004. Fuel consumed averaged 41.4 Tg y$^{-1}$, with extremes of 16.6 Tg in 2004 and 61.2 Tg in 2012. The annual rank in fuel consumed differed from burned area due to the far greater fuel loading of forests (Sect. 2.4.3). While forest comprised only 44% of burned area over the period, they accounted for 87% of fuel consumed. Average PM$_{2.5}$ emissions were 733 Gg y$^{-1}$ and, as with fuel consumed, 2004



and 2012 were the extreme years at 270 Gg and 1216 Gg, respectively. There are slight differences in the ranking of annual fuel consumption and $PM_{2.5}$ emitted resulting from the different $EFPM_{2.5}$ used for southern and western/northern forests (Table 9). Maps of annual burned area, fuel consumed, and $PM_{2.5}$ emitted averaged over 2003–2015 are shown in Figure 10. In the eastern two-thirds of the domain, fire activity and emissions are spread broadly across the southern tier while being comparatively sparse

in the north. In the west (western 11 states), fire activity has no latitudinal split, but there are large pockets where emissions are limited or absent. Much of the area in the west without emissions are in desert regions of the southwest with sparse vegetation.

The monthly distributions of burned area, fuel consumption, and $PM_{2.5}$ emitted over 2003–2015, broken down by cover type, are plotted in Figure 11. Burned area has a bimodal distribution with peaks in April and August. Summer (June, July, August) and spring (March, April, May) accounted for 49% and 31% of burned area, respectively. The ratio of herb and shrub to forest burned

area was similar for summer (1.3) and spring (1.5), but differs considerably between the peak months of April (2.7) and August (1.0). August was the most significant month for emissions, accounting for 32% $PM_{2.5}$ emitted, more than twice the share of the next highest month, which was July at 15%. While April had the third highest burned area (15% of total), it accounted for only 6.7% of $PM_{2.5}$ emitted. The geographic distribution of emissions varies considerably by season as may be seen in Figure 12.

Understanding the spatiotemporal distribution of emissions is aided by aggregating the emissions according to six regions in

Figure 13. Roughly 8% of fuel consumption and 6% of $PM_{2.5}$ emissions occurred in the winter months (Fig. 11) and were largely limited to the southeast and southcentral regions (Figs. 12). Winter $PM_{2.5}$ emissions comprised 25% and 16% of total $PM_{2.5}$ emissions in the southeast and southcentral, respectively (Fig. 14). In the southeast, 74% of winter emissions resulted from fire activity in Florida and along the gulf coast. The majority of southeast (52%) and southcentral (62%) emissions occurred in the spring. Fires in the Flint Hills region of eastern Kansas and northeast Oklahoma accounted for 44% of southcentral spring

emissions over the 13 year period. Summer was the most significant season for fuel consumption and emissions due to fire activity in the west (Fig. 12). The majority of CONUS wide fuel consumption (51%) and $PM_{2.5}$ emissions (59%) occurred during the summer. On a regional basis, southwest emissions peaked during June (46%) and during August in both the northwest (59%) and California (40%). Northwest emissions were concentrated July – September (95%), while California emissions were spread symmetrically across June–October (Fig. 14).

**3.2 Daily**

While regional level summaries on a seasonal or monthly basis are useful for understanding the general spatiotemporal distribution of wildfire emissions, daily emissions are more relevant for appreciating the potential air quality impacts of fires. For instance, US NAAQS includes a 24 h standard for $PM_{2.5}$ and an 8 h standard for $O_3$ (the latter of which can be produced through photochemical processing of VOC and NOx present in smoke plumes (Jaffe and Wigder, 2012). Wildfires are highly episodic and

even though they may persist for weeks, a significant share of a wildfire's emissions generally occur on a handful of days. For example, consider the typical large (> 2000 ha) wildfire in the west, our inventory indicates more than half its total $PM_{2.5}$ emissions occur on a single day. In the west, 1171 fires > 2000 ha in size accounted for ~85% of burned area and $PM_{2.5}$ emission from 2003– 2015. To characterize wildfire temporal intensity, emissions of $PM_{2.5}$ were summed by region for each of the 4748 days of the inventory. Figure 15 plots the fraction of regional, 2003–2015 $PM_{2.5}$ emissions released on peak emission days, the top first,

second, and fifth quantile of days. Since the north accounted for only 3% of total wildfire emissions it has been excluded from this analysis to simplify the discussion. Figure 15 shows that a small fraction of days (5%) are responsible for the majority of wildfire $PM_{2.5}$ emissions in all regions except the southeast. In fact, the percent of $PM_{2.5}$ emissions during just the top 1% of days was > 33% in California and the northwest, > 25% in the southcentral and southwest, and ~13% in the southeast. The spatiotemporal concentration of emissions is further illustrated in Figure 16, which plots the cumulative distribution of daily $PM_{2.5}$





emissions aggregated on a 10 km × 10 km grid. Five percent of the grid cell days produced 69% of total $PM_{2.5}$ emitted, and 10% of grid cell days were responsible for 82% of total $PM_{2.5}$ emitted. This analysis highlights the importance of quantifying wildfire emissions on a daily time step when assessing the potential impacts of wildfires on regional air pollution; assessments based on emissions aggregated seasonal, monthly, or even weekly may significantly understate the likelihood of acute pollution episodes.

**3.3 Comparison with non-fire emission sources**

Next we compare our wildfire $PM_{2.5}$ emissions with those from other sources as estimated in the EPA 2014 National Emission Inventory (NEI14). We focus on the west (the 11 states of the northwest, southwest, and California regions, Fig. 13a) since this region accounts for 72% of total wildfire $PM_{2.5}$ emissions (Fig. 13b) and the emissions are produced with a high temporal intensity (Figs. 14 & 15) and have resulted in severe air pollution episodes (Fann et al., 2018; Kochi et al., 2012; Rappold et al., 2014).

Non-fire $PM_{2.5}$ emission estimates for the western states were extracted from the NEI14 Tier 3 summary state level data (USEPA, 2018c). The NEI14 $PM_{2.5}$ emissions were limited to non-fire sources by excluding the Tier 3 source categories of "agricultural fires", "forest wildfires", and "prescribed burning". The NEI14 provides annual emission estimates for 2014, which are plotted with the annual sum of MFLEI $PM_{2.5}$ emissions for the west for 2003–2015, in Figure 17. The 2003–2015 annual average western wildfire $PM_{2.5}$ emitted is 525 Gg $y^{-1}$ (range 126–1034 Gg $y^{-1}$) compared with the non-fire source strength of 657 Gg $y^{-1}$ in 2014.

As discussed above, when inferring possible air quality impacts of wildfire emissions, 1 d is an appropriate time scale. Assuming the NEI14 emissions are a reasonable proxy for annual non-fire emissions across 2003–2015, and neglecting the seasonal variability of emissions, daily non-fire $PM_{2.5}$ emissions are 1.80 Gg $d^{-1}$. For all 4748 days of MFLEI period we calculated the wildfire to non-wildfire $PM_{2.5}$ emission ratio; the number of days the ratio exceeds certain thresholds is shown in Figure 18. Across the west, wildfire emissions greatly exceed non-fire sources on active fire days. On ~ 10% of days, wildfires emissions are more

than twice non-fire sources and on 60 days were >10 times non-fire sources.

   **3.4 Uncertainty**

The MFLEI pixel level best estimates of fuel consumption (FC) and emissions ($ECO_2$, $ECO$, $ECH_4$, $EPM_{2.5}$) were derived as described in Sect. 2.8 and the uncertainty in these estimates were characterized with quantiles (q=.05, .10, .25, .50, .75, .90, .95) derived from Monte Carlo style simulations (Sect. 2.9). Here we summarize the pixel level uncertainty in terms of the relative

interquartile range: RIQR = $(q_{75} - q_{25})/X$, where $q_{75}$ and $q_{25}$ are the 75% and 25% quantiles and X is the best estimate of FC or $EPM_{2.5}$; the distributions are shown in Figure 19. The mean RIQR of both FC and $EPM_{2.5}$ are ~ 67% for forest cover type and ~ 47% for herb/shrub. At the pixel level, the uncertainty is driven by the variability in fuel loading. The difference in the uncertainty estimates between forest and herb/shrub cover types results primarily from the high variability in forest fuel loading (Fig. 3). The mean RIQR are nearly 50% higher for forest compared with herb/shrub; however, the latter does have a long positive tail with ~

11% of pixels having RIQR > 90%. These high uncertainty non-forest pixels are shrub vegetation with low fuel consumption (< 350 g $m^{-2}$).

   As discussed in Sect 3.2, CONUS wildfire emissions are temporally and spatially concentrated. Considering this spatiotemporal concentration of emissions and the grid spacing typical of regional and national scale air quality modeling (4 to 12 km; USEPA, 2007; NOAA, 2018), we also estimated the uncertainty in daily MFLEI emissions aggregated on 10 km × 10 km grid

(Sect. 2.9). For purposes of air quality modeling and air regulatory activities, the uncertainty of these spatially aggregated emissions provides a more relevant metric than the pixel level uncertainty presented above. Uncertainty in the daily, aggregated FC and $PM_{2.5}$ emissions are shown in Figs. 20a-b, expressed in terms of the RIQR (calculated using the quantiles and best estimates for the spatially aggregated data). Compared with the pixel level data, the RIQR is reduced for the aggregated emissions and a



difference emerges between FC and $EPM_{2.5}$, the mean RIQR is 17% for FC and 26% for $EPM_{2.5}$. For the aggregated data we also show, in Figs. 20c-d, the distribution of relative interdecile range, RIDR = $(q_{90} – q_{10})/X$, where $q_{10}$ and $q_{90}$ are the 10% and 90% quantiles (Monte Carlo style simulations, Sect. 2.9) and X is the best estimate for FC or $EPM_{2.5}$. The mean RIDR is 32% for FC and 50% for $EPM_{2.5}$.

**3.5 Prescribed and agricultural fires**

While the focus of MFLEI is wildfires, it does include an unquantified contribution from prescribed fires – fires intentionally ignited to achieve land management objectives. The MTBS product does contain large (> 404 ha the west and > 202 ha elsewhere) prescribed fires, over 2003–2015 ~ 13% of the MTBS burned area was due to fires classified as prescribed or unknown. Additionally, the MODIS burned area product (Giglio et al., 2015) used to supplement MTBS does not distinguish between
wildfires and prescribed fires and likely includes some prescribed fire burned area. Information on prescribed fires by federal and state agencies indicate an average fire size of ~ 60 ha (NIFC, 2018). Considering the large fire focus of MTBS and the fact that prescribed fires are often low intensity understory burns, which are difficult to detect by satellite (Hawbaker et al., 2008), we believe prescribed fires account for a small share of total MFLEI emissions. Unfortunately, there is not a nationwide database that inventories prescribed fire on federal, state, and private lands. The 2015 National Prescribed Fire Use Survey Report (Melvin,
2016), based on a 2014 comprehensive survey conducted by state forestry agencies, summarizes prescribed fire activity at national and regional levels. Melvin reported CONUS prescribed fire burned area as 35,222 km$^2$ in 2014. For the same year, the MTBS prescribed fire burned area was 11,954 km$^2$ (prior to reduction for unburned to low burn severity patches as described Sect. 2.3.3), suggesting MFLEI may be be missing up to two-thirds of CONUS prescribed fire burned area. The regional summary in Melvin reports prescribed fire burned area of 25,049 km$^2$ in their southeast region (the southeast and southcentral regions used in our study,
excluding Kansas and Missouri). The 2014 MTBS data reports only 4651 km$^2$ of prescribed fire burned area for the same region, indicating most of MFLEI underrepresentation in prescribed fire emissions occurs in these southern states.

**4. Data Availability**

MFLEI is archived and publicly available at the USDA Forest Service Research Data Archive with the DOI number https://doi.org/10.2737/RDS-2017-0039.

**5. Conclusions**

We have presented the Missoula Fire Lab Wildfire Emission Inventory (MFLEI), a retrospective, wildfire emission inventory for CONUS. MFLEI was developed from multiple datasets of fire activity and burned area, a newly developed wildland fuels map and an updated emission factor database. Daily burned area was constructed using a combination of Landsat-based burn severity data (MTBS), MODIS burned area and active fire detection products, VIIRS active fire detections, incident fire perimeters, and a
spatial wildfire occurrence database. Forest fuel loading was based on a large set (> 27,000 sites) of forest inventory surface fuel measurements. Herbaceous fuel loading was estimated using site specific parameters from a soil survey database with NDVI from MODIS. Shrub fuel loading was quantified by applying numerous allometric equations linking stand structure and composition to biomass and fuels, with the structure and composition data derived from geospatial data layers of the LANDFIRE Project. MFLEI provides estimates of daily wildfire burned area, fuel consumption, and pollutant emissions at a 250 m × 250 m resolution
for 2003–2015. The inventory includes a spatially aggregated emission product (10 km × 10 km, 1 d) with uncertainty estimates to provide a more relevant representation of emission uncertainties for use in air quality modelling. MFLEI will be updated with





recent years as the MTBS data become available. The focus of MFLEI is wildfires and does not include most prescribed fire activity. In the southeast, where prescribed fire burned area is estimated to greatly exceed that of wildfires on average, the prescribed fire emissions not included in MFLEI are likely to be substantial.

MFLEI CONUS average wildfire fuel consumption and $PM_{2.5}$ emissions were estimated to be 41.4 Tg $y^{-1}$ and 733 Gg $y^{-1}$,

respectively over 2003–2015. Annual CONUS $PM_{2.5}$ emissions showed significant variability with a coefficient of variation = 0.41 and a maximum to minimum ratio of 4.5. Summer was the most active season, over half (59%) of total $PM_{2.5}$ emissions occurred in the summer (June–August), with August alone accounting for 32% of the total. Emissions were highly concentrated both temporally and spatially. Just 5% of days accounted for 57% of total $PM_{2.5}$ emitted over 2003–2015. At the spatial scale of 10 km × 10 km grid, 69% of total $PM_{2.5}$ originated from 5% of grid cell days with fire activity. Fires in the west (western 11 states)

accounted for 56% of burned area, 60% of fuel consumption, and 72% of $PM_{2.5}$ emitted over 2003–2015. The southeast and south central regions were largely responsible for the balance of burned area and emissions. The northern tier states across central and eastern CONUS produced < 3 % of total $PM_{2.5}$ emissions. In the west, wildfire $PM_{2.5}$ emissions dwarfed those from non-fire sources during active fire periods. Comparison of MFLEI $PM_{2.5}$ emissions with the EPA 2014 National Emission Inventory indicated that in the west, wildfires exceeded all non-fire primary sources of $PM_{2.5}$ by a factor of > 5 on nearly 200 days over

2003–2015. Quantified with the relative interdecile range, the uncertainties in daily fuel consumption and $PM_{2.5}$ emissions, at the spatial scale of 10 km × 10 km, were estimated to be 32% and 50% respectively.

**Appendix A**

**Table A1.** Average small fire duration based the fire discovery and containment dates from years 2003–2015 of the Fire Occurrence Database (Short, 2017).

**Appendix B**

**Table B1.** Available canopy fuel (ACF) best estimates (see Sect. 2.4.1) and optimized parameters for Weibull pdf fits. Parameters predict ACF in units of ton $acre^{-1}$

**Appendix C**

This appendix demonstrates the four step process used to estimate shrub fuel loading. Three LF existing vegetation products are

used EVT, EVC, and EVH. The height estimates at each pixel in the EVH product are thematic classes representing a range of potential heights (Table C1) which enables three values of shrub fuels to be estimated at each pixel (average, maximum, and minimum). Likewise, the EVC product is thematic classes which providing a 10 percentage point range in potential vegetation cover (Table C2). However, the shrub fuel loading calculation simply uses the median value vegetation cover range. To illustrate, consider a pixel with an EVT of class of Big Sagebrush shrubland, an EVH class of 105, and an EVC class of 112 the fuel loading

proceed as follows:

First, the crown volume is derived from the three EVH estimates (0.5, 0.75, and 1.0 m) as the product of these EVH values and the projected crown area on a horizontal surface (PCH), the latter of which is estimated using Eq. C1 (Frandsen, 1983):

$$log_{10}(\text{PCH}) = -0.8471 + 2.2953 log_{10}(HT), \tag{C1}$$





where PCH is the projected horizontal crown area in cm$^2$ and HT is the estimated shrub height in cm (from the EVH product). Per stem above ground biomass estimates are then derived from the crown volume estimates using an allometric equation for Sagebrush shrubs from the RVS allometry library:

$$PSB = 201.4062 + 1.162 \times \text{VOL}, \tag{C2}$$

where PSB is per stem biomass (g stem$^{-1}$) and VOL is crown volume (dm$^3$). Next, the pixel stem density, SD, (stem ha$^{-1}$) is estimated to expand PSB to a per-area basis:

$$SD = \left(\frac{1.0e8}{\text{PCH}}\right) \times \text{CC}, \tag{C3}$$

where SD is stem density, CC is the fractional canopy cover from EVC (Table C2), and the value 1.0e8 converts cm$^2$ to a per
hectare basis. The total shrub biomass (TSB; kg ha$^{-1}$) is the product of PSB and SD. Figure C1 shows the TSB estimates for the pixel used in this example. This process was conducted at each pixel with a shrub EVT using the range of heights from EVH to provide lower, upper, and middle estimates of fuel loading. The allometric equation used to estimate PSB depends on the pixel EVT and is selected from 31 available in the RVS allometry library.

Supplement link provided by journal.

Author contributions. SPU and WMH designed the inventory model and selected the fire activity, land cover, and fuel loading datasets used as inventory input. MCR conceived and produced the rangeland fuel loading dataset and created the final land cover
map. RPS processed the FIA data to provide the plot level datasets of surface fuel loading and processed the forest understory carbon dataset used to derive understory fuel loading. REC processed the activity data to create the burned area maps. SPU was responsible for finalizing the burn date assignment, developing forest fuel loading classifications, selecting fuel condition and fuel consumption methodologies, and compiling emission factors. SPU conducted all final inventory calculations and devised and implemented the uncertainty analysis. SPU prepared the paper with contributions from MCR and was responsible for preparing
the inventory dataset published in the USDA Forest Service Research Data Archive.

Competing interests. The authors declare that they have no conflict of interest.

Acknowledgements. This work was financially supported by the USDA Forest Service Rocky Mountain Research Station, Joint Fire Sciences Program grant No. 12-1-07-1, and NOAA ESS – Atmospheric Chemistry, Carbon Cycle, and Climate (AC4), Grant No. NA13OAR4310087.



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





**Tables**

**Table 1.** MFLEI cover types and fuel codes

| Cover type code | Fuel code | Cover type | Generalized cover type |
|---|---|---|---|
| -99 | 0 | Non-fuel | Non-fuel |
| 1 | 1 | Herbaceous | Herbaceous |
| 2 | 2 | Shrub / scrub | Shrub |
| 100 | 1100 | White / red / jack pine group | Northern conifer |
| 120 | 1120 | Spruce / fir group | Northern conifer |
| 140 | 1140 | Longleaf / slash pine group | Southern conifer |
| 160 | 1160 | Loblolly / shortleaf pine group | Southern conifer |
| 180 | 1180 /2180 | Pinyon / juniper group | Pinyon juniper |
| 200 | 1200 | Douglas-fir group | Western conifer / softwood |
| 220 | 1220 | Ponderosa pine group | Western conifer / softwood |
| 240 | 1240 | Western white pine group | Western conifer / softwood |
| 260 | 1260 | Fir / spruce / mountain hemlock group | Western conifer / softwood |
| 280 | 1280 | Lodgepole pine group | Western conifer / softwood |
| 300 | 1300 | Hemlock / Sitka spruce group | Western conifer / softwood |
| 320 | 1320 | Western larch group | Western conifer / softwood |
| 340 | 1340 | Redwood group | Western conifer / softwood |
| 360 | 1360 | Other western softwoods group | Western conifer / softwood |
| 370 | 1370 | California mixed conifer group | Western conifer / softwood |
| 380 | 1380 | Exotic softwoods group | Western conifer / softwood |
| 400 | 1400 | Oak / pine group | Hardwood |
| 500 | 1500 | Oak / hickory group | Hardwood |
| 600 | 1600 | Oak / gum / cypress group | Hardwood |
| 700 | 1700/2700 | Elm / ash / cottonwood group | Hardwood |
| 800 | 1800 | Maple / beech / birch group | Hardwood |
| 900 | 1900/2900 | Aspen / birch group | Western hardwood |
| 910 | 1910 | Alder / maple group | Western hardwood |
| 920 | 1920 | Western oak group | Western hardwood |
| 940 | 1940 | Tanoak / laurel group | Western hardwood |
| 950 | 1950/2950 | Other western hardwoods group | Western hardwood |
| 980 | 1980 | Tropical hardwoods group | Western hardwood |
| 990 | 1990 | Exotic hardwoods group | Western hardwood |



**Table 2.** MTBS burn severity class percent distribution by generalized cover types for 2003-2013.

| Generalized Cover Type | BSEV =1 | BSEV = 2 | BSEV = 3 | BSEV = 4 |
|---|---|---|---|---|
| Herbaceous | 18 | 68 | 11 | 3 |
| Shrub | 17 | 57 | 22 | 4 |
| Northern conifer | 18 | 34 | 19 | 29 |
| Southern conifer | 25 | 61 | 12 | 2 |
| Pinyon juniper | 24 | 43 | 25 | 8 |
| Western conifer / softwood | 25 | 32 | 22 | 21 |
| Hardwood | 27 | 62 | 10 | 1 |
| Western hardwood | 18 | 38 | 25 | 19 |




**Table 3.** Description of fuel components

| General fuel type | Fuel component | Strata | Description |
|---|---|---|---|
| Litter | Litter | Surface | Loose, freshly fallen plant material found on the top surface of the forest floor which includes needles, leaves, cones, and dead herbaceous stems.[1] |
| Duff | Duff | Surface | Layer just below the litter consisting of partially decomposed biomass whose origins cannot be determined.[1] |
| Down dead wood | 1 h (small woody) | Surface | < 1 cm diameter |
| | 10 h (medium woody) | Surface | 1-2.5 cm diameter |
| | 100 h (large woody) | Surface | 2.5-7.6 cm diameter |
| | s3to9 (coarse woody debris) | Surface | Sound[2] logs 7.6-22.9 cm diameter |
| | s9to20 (coarse woody debris) | Surface | Sound[2] logs 22.9-50.8 cm diameter |
| | sgt20 (coarse woody debris) | Surface | Sound[2] logs >50.8 cm diameter |
| | r3to9 (coarse woody debris) | Surface | Rotten[2] logs 7.6-22.9 cm diameter |
| | r9to20 (coarse woody debris) | Surface | Rotten[2] logs 22.9-50.8 cm diameter |
| | rgt20r (coarse woody debris) | Surface | Rotten[2] logs >50.8 cm diameter |
| Herb | Herb | Understory | Herbs (above ground portion) |
| Shrub | Shrub | Understory | Woody shrubs (above ground portion) |
| Canopy | Available Canopy Fuel (ACF) | Canopy | Foliage and twigs ≤ 6 mm diameter |

[1]O'Connell et al. (2016)

[2]Sound logs are logs assigned FIA decay classes 1, 2, or 3 and rotten logs are logs assigned FIA decay class 4 or 5 (O'Connell et al., 2016)

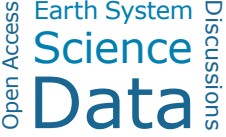

**Table 4.** Fuel loading (kg m⁻²) by fuel component for the Fuel Type Group (FTG) Classification. Litter and duff depth in cm. See Table 3 for descriptions.

| Fuel Code | FTG Code | No. Plots | Litter | Litter depth | 1 hr | 10 hr | 100 hr | s3to9 | s9to20 | sgt20 | r3to9 | r9to20 | rgt20 | Duff | Duff depth | Herb | Shrub | ACF |
|---|---|---|---|---|---|---|---|---|---|---|---|---|---|---|---|---|---|---|
| 1100 | 100 | 45 | 2.34 | 0.44 | 0.03 | 0.14 | 0.36 | 0.38 | 0.21 | 0.05 | 0.04 | 0.04 | 0.00 | 4.12 | 0.30 | 0.06 | 0.23 | 0.50 |
| 1120 | 120 | 100 | 0.65 | 0.34 | 0.02 | 0.10 | 0.32 | 0.48 | 0.18 | 0.01 | 0.10 | 0.05 | 0.00 | 10.21 | 2.08 | 0.03 | 0.28 | 0.91 |
| 1140 | 140 | 79 | 3.14 | 0.59 | 0.01 | 0.09 | 0.17 | 0.11 | 0.04 | 0.00 | 0.02 | 0.01 | 0.00 | 3.26 | 0.24 | 0.05 | 0.47 | 0.29 |
| 1160 | 160 | 266 | 3.06 | 0.56 | 0.01 | 0.10 | 0.28 | 0.12 | 0.06 | 0.00 | 0.05 | 0.02 | 0.00 | 2.64 | 0.19 | 0.03 | 0.49 | 0.16 |
| 1180 | 180 | 5626 | 0.43 | 0.12 | 0.02 | 0.06 | 0.16 | 0.10 | 0.07 | 0.01 | 0.02 | 0.03 | 0.00 | 0.38 | 0.04 | 0.10 | 0.31 | 0.68 |
| 1200 | 200 | 3558 | 0.64 | 0.34 | 0.04 | 0.14 | 0.50 | 0.43 | 0.50 | 0.45 | 0.10 | 0.29 | 0.28 | 1.39 | 0.28 | 0.07 | 0.42 | 1.18 |
| 1220 | 220 | 2163 | 1.19 | 0.34 | 0.01 | 0.09 | 0.26 | 0.26 | 0.24 | 0.11 | 0.04 | 0.09 | 0.08 | 1.30 | 0.15 | 0.10 | 0.30 | 0.46 |
| 1240 | 240 | 30 | 0.78 | 0.23 | 0.01 | 0.06 | 0.19 | 0.25 | 0.42 | 0.56 | 0.07 | 0.11 | 0.09 | 1.33 | 0.15 | 0.24 | 0.36 | 0.41 |
| 1260 | 260 | 3000 | 0.53 | 0.27 | 0.03 | 0.12 | 0.42 | 0.58 | 0.84 | 0.31 | 0.15 | 0.41 | 0.22 | 1.74 | 0.35 | 0.06 | 0.36 | 1.23 |
| 1280 | 280 | 1334 | 0.86 | 0.25 | 0.02 | 0.09 | 0.38 | 0.89 | 0.49 | 0.06 | 0.18 | 0.24 | 0.07 | 2.22 | 0.25 | 0.20 | 0.20 | 0.67 |
| 1300 | 300 | 521 | 0.66 | 0.35 | 0.03 | 0.16 | 0.53 | 0.67 | 1.35 | 1.88 | 0.18 | 0.71 | 0.94 | 2.66 | 0.54 | 0.25 | 0.31 | 1.38 |
| 1320 | 320 | 159 | 1.37 | 0.40 | 0.04 | 0.17 | 0.65 | 1.24 | 0.90 | 0.17 | 0.14 | 0.35 | 0.05 | 3.29 | 0.37 | 0.46 | 0.05 | 0.63 |
| 1340 | 340 | 63 | 3.00 | 0.87 | 0.03 | 0.16 | 0.52 | 0.55 | 0.74 | 2.03 | 0.12 | 0.37 | 0.71 | 2.73 | 0.31 | 0.09 | 0.49 | 1.66 |
| 1360 | 360 | 796 | 0.52 | 0.15 | 0.01 | 0.05 | 0.15 | 0.15 | 0.17 | 0.05 | 0.03 | 0.07 | 0.05 | 0.68 | 0.08 | 0.13 | 0.20 | 0.39 |
| 1370 | 370 | 894 | 1.70 | 0.49 | 0.02 | 0.15 | 0.43 | 0.40 | 0.48 | 0.60 | 0.06 | 0.21 | 0.37 | 1.85 | 0.21 | 0.14 | 0.32 | 1.04 |
| 1400 | 400 | 133 | 2.68 | 0.49 | 0.01 | 0.12 | 0.43 | 0.22 | 0.12 | 0.00 | 0.06 | 0.04 | 0.00 | 2.58 | 0.18 | 0.05 | 0.42 | 0.13 |
| 1500 | 500 | 1375 | 1.79 | 0.39 | 0.01 | 0.09 | 0.32 | 0.24 | 0.13 | 0.01 | 0.04 | 0.03 | 0.01 | 1.22 | 0.10 | 0.00 | 0.37 | 0.06 |
| 1600 | 600 | 129 | 1.37 | 0.30 | 0.01 | 0.10 | 0.29 | 0.15 | 0.15 | 0.03 | 0.04 | 0.09 | 0.00 | 3.28 | 0.28 | 0.00 | 0.39 | 0.33 |
| 1700 | 700 | 50 | 1.14 | 0.36 | 0.03 | 0.15 | 0.85 | 0.32 | 0.39 | 0.04 | 0.04 | 0.12 | 0.12 | 2.40 | 0.29 | 0.01 | 0.25 | 0.22 |
| 1800 | 800 | 336 | 1.97 | 0.44 | 0.02 | 0.14 | 0.44 | 0.54 | 0.37 | 0.06 | 0.09 | 0.09 | 0.00 | 3.39 | 0.29 | 0.07 | 0.22 | 0.21 |
| 1900 | 900 | 619 | 1.25 | 0.25 | 0.02 | 0.09 | 0.47 | 0.47 | 0.33 | 0.04 | 0.11 | 0.13 | 0.03 | 3.31 | 0.26 | 0.03 | 0.30 | 0.34 |
| 1910 | 910 | 222 | 2.07 | 0.46 | 0.02 | 0.14 | 0.47 | 0.38 | 0.59 | 0.95 | 0.08 | 0.26 | 0.48 | 3.33 | 0.29 | 0.05 | 0.47 | 0.40 |
| 1920 | 920 | 907 | 1.73 | 0.37 | 0.02 | 0.10 | 0.28 | 0.21 | 0.17 | 0.11 | 0.03 | 0.06 | 0.05 | 0.99 | 0.08 | 0.10 | 0.40 | 0.24 |
| 1940 | 940 | 263 | 2.23 | 0.48 | 0.03 | 0.16 | 0.49 | 0.46 | 0.44 | 0.62 | 0.06 | 0.20 | 0.32 | 2.53 | 0.21 | 0.09 | 0.54 | 0.48 |
| 1950 | 950 | 1590 | 0.93 | 0.20 | 0.02 | 0.07 | 0.17 | 0.11 | 0.09 | 0.04 | 0.02 | 0.03 | 0.01 | 0.94 | 0.08 | 0.06 | 0.31 | 0.41 |
| 2180 | 180 | 759 | 0.45 | 0.13 | 0.00 | 0.03 | 0.13 | 0.02 | 0.00 | 0.00 | 0.01 | 0.00 | 0.00 | 0.30 | 0.03 | 0.10 | 0.31 | 0.40 |
| 2700 | 700 | 202 | 0.79 | 0.25 | 0.01 | 0.11 | 0.37 | 0.21 | 0.12 | 0.04 | 0.04 | 0.05 | 0.00 | 0.47 | 0.06 | 0.01 | 0.25 | 0.17 |
| 2900 | 900 | 92 | 2.39 | 0.49 | 0.01 | 0.11 | 0.31 | 0.39 | 0.20 | 0.01 | 0.10 | 0.04 | 0.00 | 3.59 | 0.28 | 0.03 | 0.30 | 0.18 |
| 2950 | 950 | 1813 | 0.47 | 0.10 | 0.01 | 0.05 | 0.13 | 0.02 | 0.00 | 0.00 | 0.00 | 0.00 | 0.00 | 0.09 | 0.01 | 0.06 | 0.31 | 0.13 |



**Table 5**. Fuel moisture regimes used for simulating fuel consumption.

| Regime | NFDRS station data moisture content range | Moisture content used in fuel consumption simulations | |
|---|---|---|---|
| | 1000 h (%) | 1000 h (%) | duff (%) |
| very dry | <= 10 | 10 | 20 |
| dry | > 10 and <= 25 | 20 | 40 |
| moderate | > 25 and <= 35 | 30 | 60 |
| moist | > 35 | 40 | 80 |



**Table 6.** Best estimates and ranges of the combustion completeness by fuel component according to moisture regime and forest type group. Best estimates are based on cited references. Low and high ranges assigned as approximately +/- 20 %.

| Fuel component[1] | Moisture regime | | | | | | | | | | | | Reference |
| | Very dry | | | Dry | | | Moderate | | | Moist | | | |
| | best estimate | low | high | best estimate | Low | high | best estimate | low | high | best estimate | low | high | |
|---|---|---|---|---|---|---|---|---|---|---|---|---|---|
| colspan | Western and Northern Forest Type Groups (All forests EXCEPT Fuel Codes 1140,1160,1400,1500, and 1600) | | | | | | | | | | | | |
| Shrub | 0.90 | 0.80 | 1.00 | 0.90 | 0.80 | 1.00 | 0.90 | 0.80 | 1.00 | 0.90 | 0.80 | 1.00 | a |
| Herb | 0.90 | 0.80 | 1.00 | 0.90 | 0.80 | 1.00 | 0.90 | 0.80 | 1.00 | 0.90 | 0.80 | 1.00 | b |
| HR1 | 0.95 | 0.90 | 1.00 | 0.95 | 0.90 | 1.00 | 0.95 | 0.90 | 1.00 | 0.95 | 0.90 | 1.00 | c |
| HR10 | 0.86 | 0.72 | 1.00 | 0.86 | 0.72 | 1.00 | 0.86 | 0.72 | 1.00 | 0.86 | 0.72 | 1.00 | c |
| HR100 | 0.78 | 0.62 | 0.94 | 0.78 | 0.62 | 0.94 | 0.78 | 0.62 | 0.94 | 0.78 | 0.62 | 0.94 | c |
| Litter | 0.90 | 0.80 | 1.00 | 0.90 | 0.80 | 1.00 | 0.90 | 0.80 | 1.00 | 0.90 | 0.80 | 1.00 | d |
| Duff | 0.75 | 0.60 | 0.90 | 0.67 | 0.54 | 0.80 | 0.58 | 0.46 | 0.70 | 0.50 | 0.40 | 0.60 | e |
| s3to9 | 0.93 | 0.86 | 1.00 | 0.88 | 0.76 | 1.00 | 0.81 | 0.65 | 0.97 | 0.71 | 0.56 | 0.85 | c |
| s9to20 | 0.60 | 0.48 | 0.72 | 0.50 | 0.40 | 0.60 | 0.41 | 0.33 | 0.49 | 0.32 | 0.25 | 0.38 | c |
| sgt20 | 0.50 | 0.40 | 0.60 | 0.41 | 0.32 | 0.49 | 0.32 | 0.25 | 0.38 | 0.24 | 0.19 | 0.29 | c |
| r3to9 | 0.96 | 0.92 | 1.00 | 0.88 | 0.76 | 1.00 | 0.70 | 0.56 | 0.84 | 0.43 | 0.34 | 0.52 | c |
| r9to20 | 0.78 | 0.62 | 0.94 | 0.59 | 0.47 | 0.71 | 0.38 | 0.30 | 0.46 | 0.20 | 0.16 | 0.24 | c |
| rgt20 | 0.57 | 0.46 | 0.68 | 0.43 | 0.34 | 0.52 | 0.31 | 0.25 | 0.37 | 0.21 | 0.17 | 0.25 | c |
| colspan | Southern Forest Type Groups (Fuel Codes 1140,1160,1400,1500, and 1600) | | | | | | | | | | | | |
| Shrub | 0.90 | 0.80 | 1.00 | 0.90 | 0.80 | 1.00 | 0.90 | 0.80 | 1.00 | 0.90 | 0.80 | 1.00 | a |
| Herb | 0.90 | 0.80 | 1.00 | 0.90 | 0.80 | 1.00 | 0.90 | 0.80 | 1.00 | 0.90 | 0.80 | 1.00 | b |
| HR1 | 0.95 | 0.90 | 1.00 | 0.95 | 0.90 | 1.00 | 0.95 | 0.90 | 1.00 | 0.95 | 0.90 | 1.00 | f |
| HR10 | 0.86 | 0.72 | 1.00 | 0.86 | 0.72 | 1.00 | 0.86 | 0.72 | 1.00 | 0.86 | 0.72 | 1.00 | f |
| HR100 | 0.40 | 0.32 | 0.48 | 0.40 | 0.32 | 0.48 | 0.40 | 0.32 | 0.48 | 0.40 | 0.32 | 0.48 | f |
| Litter | 0.90 | 0.80 | 1.00 | 0.90 | 0.80 | 1.00 | 0.90 | 0.80 | 1.00 | 0.90 | 0.80 | 1.00 | d |
| Duff | 0.15 | 0.12 | 0.18 | 0.10 | 0.08 | 0.12 | 0.05 | 0.00 | 0.10 | 0.05 | 0.00 | 0.10 | g |
| s3to9 | 0.33 | 0.26 | 0.40 | 0.18 | 0.14 | 0.22 | 0.10 | 0.08 | 0.12 | 0.05 | 0.04 | 0.06 | f |
| s9to20 | 0.33 | 0.26 | 0.40 | 0.18 | 0.14 | 0.22 | 0.10 | 0.08 | 0.12 | 0.05 | 0.04 | 0.06 | f |
| sgt20 | 0.33 | 0.26 | 0.40 | 0.18 | 0.14 | 0.22 | 0.10 | 0.08 | 0.12 | 0.05 | 0.04 | 0.06 | f |
| r3to9 | 0.41 | 0.33 | 0.49 | 0.27 | 0.22 | 0.32 | 0.11 | 0.09 | 0.13 | 0.05 | 0.04 | 0.06 | f |
| r9to20 | 0.41 | 0.33 | 0.49 | 0.27 | 0.22 | 0.32 | 0.11 | 0.09 | 0.13 | 0.05 | 0.04 | 0.06 | f |
| rgt20 | 0.41 | 0.33 | 0.49 | 0.27 | 0.22 | 0.32 | 0.11 | 0.09 | 0.13 | 0.05 | 0.04 | 0.06 | f |
| colspan | Rangeland | | | | | | | | | | | | |
| Herb | 0.93 | 0.86 | 1.00 | 0.93 | 0.86 | 1.00 | 0.93 | 0.86 | 1.00 | 0.93 | 0.86 | 1.00 | b |
| Shrub | 0.90 | 0.80 | 1.00 | 0.90 | 0.80 | 1.00 | 0.90 | 0.80 | 1.00 | 0.90 | 0.80 | 1.00 | a |

[1]See Table 3 for description

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



**Table 7.** Fraction of forest canopy consumed according to burn severity classification. After Miller and Yool (2003).

| Burn Severity Code | Burn Severity Thematic Class | Fraction of canopy consumed | | |
| --- | --- | --- | --- | --- |
| | | Best estimate | Lower range | Upper range |
| 1 | Unburned to low severity | 0 | 0 | 0 |
| 2 | Low severity | 0.125 | 0.05 | 0.20 |
| 3 | Moderate severity | 0.60 | 0.50 | 0.70 |
| 4 | High severity | 1 | 1 | 1 |





**Table 8.** Statistics for the linear regression of EF as a function of MCE for field data from 78 forest fires and 20 rangeland fires (Table S4 and Figs. S1 and S2)

| | Intercept | Slope | $R^2$ | Standard Error |
|---|---|---|---|---|
| | | Forest | | |
| $EFCO_2$ | -476 | 2304 | 0.87 | 23 |
| EFCO | 1088 | -1084 | 0.99 | 2.5 |
| $EFCH_4$ | 96.2 | -100.7 | 0.79 | 1.4 |
| $EFPM_{2.5}$ | 209.0 | -211.3 | 0.53 | 4.9 |
| | | Rangeland | | |
| $EFCO_2$ | -673 | 2505 | 0.89 | 17 |
| EFCO | 1105 | -1103 | 1.00 | 1 |
| $EFCH_4$ | 62.9 | -64.2 | 0.79 | 0.6 |
| $EFPM_{2.5}$ | 76 | -70.1 | 0.07 | 4.8 |



**Table 9.** Best estimate MCE and EF (g kg$^{-1}$) for generalized fire types from multiple field studies. The standard deviation for MCE is provided in parentheses. EF are based on the linear fits in Table 8 at the fire type average MCE value. The MCE values are from Urbanski (2014).

| General Fuel Type | MCE | EFCO$_2$ | EFCO | EFCH$_4$ | EFPM$_{2.5}$ |
|---|---|---|---|---|---|
| Southern Forests[1] | 0.933 (0.013) | 1674 | 77 | 2.5 | 11.9 |
| Western & Northern Forests[2] | 0.881 (0.031) | 1554 | 133 | 7.5 | 22.8 |
| Rangeland[3] | 0.938 (0.020) | 1677 | 70 | 2.7 | 10.2 |

[1]Fuel codes 1140, 1160, 1400, 1500, and 1600

[2]All forest fuel codes except 1140, 1160, 1400, 1500, and 1600

[3]Fuel codes 1 and 2




**Table 10.** Sample generation methods employed in Monte Carlo style simulation of emission intensity uncertainty

| Fuel Component | Sample Generation | Details |
|---|---|---|
| Surface Fuel Loading | Sampling of surface fuel data from FIA plots | Supplemental dataset Fuel_Load_Plot_Data.csv |
| Understory Fuel Loading | Empirical distribution | Supplemental dataset Understory_Fuel_Dist.csv |
| Available Canopy Fuel | Weibull distribution | Table B4 |
| Herbaceous Fuel Loading | Normal Distribution | See text |
| Shrub Fuel Loading | Normal Distribution | See text |
| Fraction of Fuel Consumed | Uniform distribution | Table 6 and Table 7 |
| Emission Factors | Normal or truncated normal distribution | Table 8 and Table 9. See text. |





**Table A1.** Average small fire duration based the fire discovery and containment dates from years 2003 – 2015 of the Fire Occurrence Database (Short, 2017).

| Fire size (ha) | Number of fires | Average duration (d) | Standard Deviation of duration (d) |
|---|---|---|---|
| 0 – 31 | 43915 | 2 | 7 |
| 31 – 62 | 6894 | 4 | 12 |
| 62 – 94 | 2673 | 5 | 13 |
| 94 – 125 | 1704 | 7 | 16 |
| 125 – 156 | 962 | 8 | 17 |
| 156 – 188 | 838 | 8 | 18 |
| 188 – 625 | 216 | 9 | 17 |



**Table B1.** Available canopy fuel (ACF) best estimates (see Sect. 2.4.1) and optimized parameters for Weibull PDF fits. Parameters predict ACF in units of ton acre[-1].

| Fuel Code | N[1] | Best est. ACF (kg m[-2] ) | Best est. ACF (ton acre[-1]) | Shape parameter | Scale parameter |
|---|---|---|---|---|---|
| 1100 | 12199 | 0.50 | 2.23 | 1.73 | 2.48 |
| 1120 | 21990 | 0.91 | 4.05 | 1.33 | 4.40 |
| 1140 | 21443 | 0.29 | 1.28 | 1.30 | 1.38 |
| 1160 | 61276 | 0.16 | 0.71 | 1.33 | 0.77 |
| 1170 | 1582 | 0.36 | 1.60 | 1.55 | 1.77 |
| 1180 | 19045 | 0.68 | 3.02 | 1.56 | 3.37 |
| 1200 | 15112 | 1.18 | 5.25 | 1.70 | 5.85 |
| 1220 | 9554 | 0.46 | 2.06 | 1.66 | 2.30 |
| 1240 | 90 | 0.41 | 1.85 | 1.54 | 2.04 |
| 1260 | 10886 | 1.23 | 5.48 | 1.80 | 6.14 |
| 1280 | 5425 | 0.67 | 2.99 | 1.59 | 3.32 |
| 1300 | 2002 | 1.38 | 6.16 | 1.56 | 6.82 |
| 1320 | 684 | 0.63 | 2.80 | 1.48 | 3.07 |
| 1340 | 239 | 1.66 | 7.39 | 2.29 | 8.29 |
| 1360 | 2573 | 0.39 | 1.75 | 1.19 | 1.86 |
| 1370 | 2173 | 1.04 | 4.62 | 1.91 | 5.19 |
| 1380 | 543 | 0.63 | 2.80 | 1.01 | 2.82 |
| 1400 | 34528 | 0.13 | 0.57 | 1.03 | 0.57 |
| 1500 | 58266 | 0.06 | 0.25 | 0.85 | 0.23 |
| 1600 | 17157 | 0.33 | 1.48 | 0.62 | 0.98 |
| 1700 | 134 | 0.22 | 0.97 | 0.91 | 0.93 |
| 1800 | 25727 | 0.21 | 0.92 | 0.91 | 0.88 |
| 1900 | 1736 | 0.34 | 1.50 | 1.03 | 1.51 |
| 1910 | 945 | 0.40 | 1.76 | 1.02 | 1.78 |
| 1920 | 1564 | 0.24 | 1.05 | 0.80 | 0.93 |
| 1940 | 774 | 0.48 | 2.13 | 1.19 | 2.26 |
| 1950 | 294 | 0.41 | 1.81 | 1.02 | 1.82 |
| 1970 | 2119 | 0.18 | 0.82 | 1.05 | 0.83 |
| 1980 | 166 | 0.10 | 0.43 | 1.04 | 0.44 |
| 1990[1] | 0 | 0.10 | 0.43 | 1.04 | 0.44 |
| 2180 | 1257 | 0.40 | 1.79 | 1.36 | 1.96 |
| 2700 | 6859 | 0.17 | 0.77 | 0.82 | 0.69 |
| 2900 | 18279 | 0.18 | 0.79 | 0.92 | 0.75 |
| 2950 | 690 | 0.13 | 0.57 | 0.91 | 0.55 |

[1]N = number of FIA plots used in deriving best estimate ACF and Weibull PDF fits.

[2]Values for fuel code 1990 set to those of fuel code 1980 due to lack of data





**Table C1.** Thematic classes representing shrub heights in the Landfire EVH product and the associated height values represented in the RVS fuel modelling system.

| EVH Class Code | EVH Classes | RVS shrub height (m) | | |
|---|---|---|---|---|
| | | Minimum | Median | Maximum |
| 104 | Shrub height 0 to 0.5 m | 0.1 | 0.25 | 0.5 |
| 105 | Shrub height 0.5 to 1.0 m | 0.5 | 0.75 | 1 |
| 106 | Shrub height 1.0 to 3.0 m | 1 | 2 | 3 |
| 107 | Shrub height > 3.0 m | 3 | 4 | 5 |



**Table C2.** Thematic classes representing shrub canopy in the Landfire EVC product and the associated canopy cover used in the RVS fuel modelling system.

| EVC Class Code | EVC Classes | RVS canopy cover |
|---|---|---|
| | | % |
| 111 | Shrub cover >= 10 and < 20 | 15 |
| 112 | Shrub cover >= 20 and < 30 | 25 |
| 113 | Shrub cover >= 30 and < 40 | 35 |
| 114 | Shrub cover >= 40 and < 50 | 45 |
| 115 | Shrub cover >= 50 and < 60 | 55 |
| 116 | Shrub cover >= 60 and < 70 | 65 |
| 117 | Shrub cover >= 70 and < 80 | 75 |
| 118 | Shrub cover >= 80 and < 90 | 85 |
| 119 | Shrub cover >= 90 and <= 100 | 95 |



**Figure captions**

Figure 1. MFLEI land cover type map. White regions are non-fuel cover type. Cover type codes are described in Table 1.

Figure 2. Location of FIA plots used to develop surface fuel loading classifications.

Figure 3. The distribution of surface fuel loading for the FIA plots of three FTG, Loblolly/shortleaf pine (160), Douglas-fir (200), and California mixed conifer (370).

Figure 4. Best estimate (Table 4) forest fuel loading in canopy, understory, and surface fuels by fuel type.

Figure 5. Fraction of best estimate (Table 4) total forest fuel loading in surface fuel loading groups by fuel type.

Figure 6. Abbreviated flow of data and actions in RVS to produce rangeland fuel loadings. EVT, EVC and EVH are Existing Vegetation Type, Existing Vegetation Cover, and Existing Vegetation Height from the Landfire Project.

Figure 7. Relationship between annual production and annual maximum NDVI on 51 grassland vegetation types.

Figure 8. Map of best estimate fuel loading for forest and rangelands in g m$^{-2}$.

Figure 9. Annual burned area, fuel consumed, and PM$_{2.5}$ emitted for 2003-2015.

Figure 10. Annual burned area, fuel consumed, and PM$_{2.5}$ emitted averaged over 2003-2015.

Figure 11. Monthly distributions of burned area, fuel consumption, and PM$_{2.5}$ emitted over 2003-2015, broken down by cover type.

Figure 12. Seasonal PM$_{2.5}$ emitted average over 2003-2015.

Figure 13. Top panel: geographic regions. Bottom panel: Burned area, fuel consumption, and PM$_{2.5}$ emitted by region.

Figure 14. Monthly PM$_{2.5}$ emitted averaged over 2003-2015.

Figure 15. Fraction of regional, 2003-2015 PM$_{2.5}$ emissions released on peak days.

Figure 16. Cumulative distribution of daily PM$_{2.5}$ emissions aggregated on a 10 km × 10 km grid. Dashed line and dashed – dotted line mark 5% and 10% of grid cell days with emissions.

Figure 17. Annual PM$_{2.5}$ emitted in the west.

Figure 18. Number of days over 2003-2015 when the wildfire to non-wildfire PM$_{2.5}$ emission ratio in the west exceeds thresholds of 2, 5, 10, 15, and 20.

Figure 19. Distribution of relative interquartile range from pixel level Monte Carlo style simulations.

Figure 20. Distribution of relative interquartile range (top panels) and relative interdecile range (bottom panels) from 10 km × 10 km gridded Monte Carlo style simulations.

Figure A1. The burn day distribution for the 12,500–25,000 ha size class. Distributions for all six size classes are provided in the dataset supplement (file\Supplements\BurnDayDist.csv).

Figure C1. Total shrub biomass estimates for a pixel with EVT class of Big Sagebrush shrubland, EVH class of 105, and EVC class of 112 (see text).



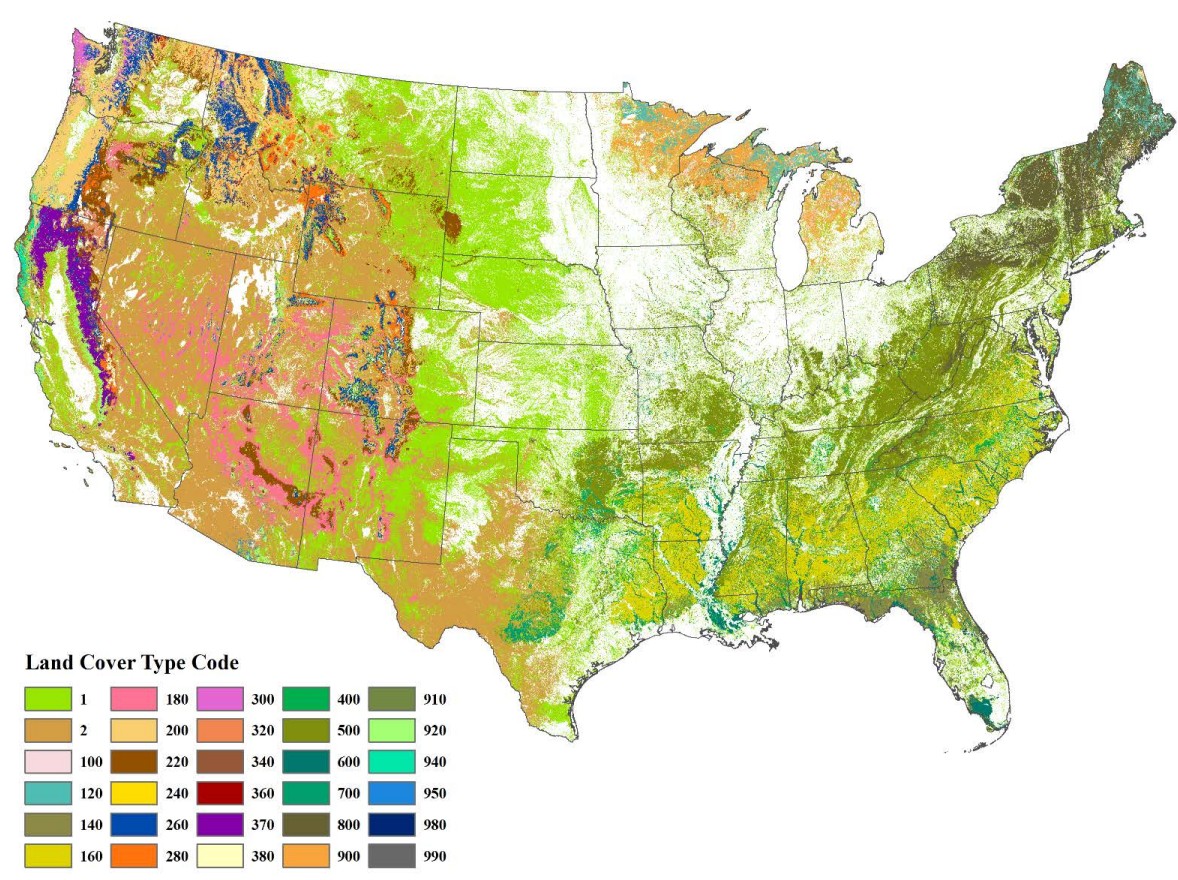

Figure 1. MFLEI land cover type map. White regions are non-fuel cover type. Cover type codes are described in Table 1.



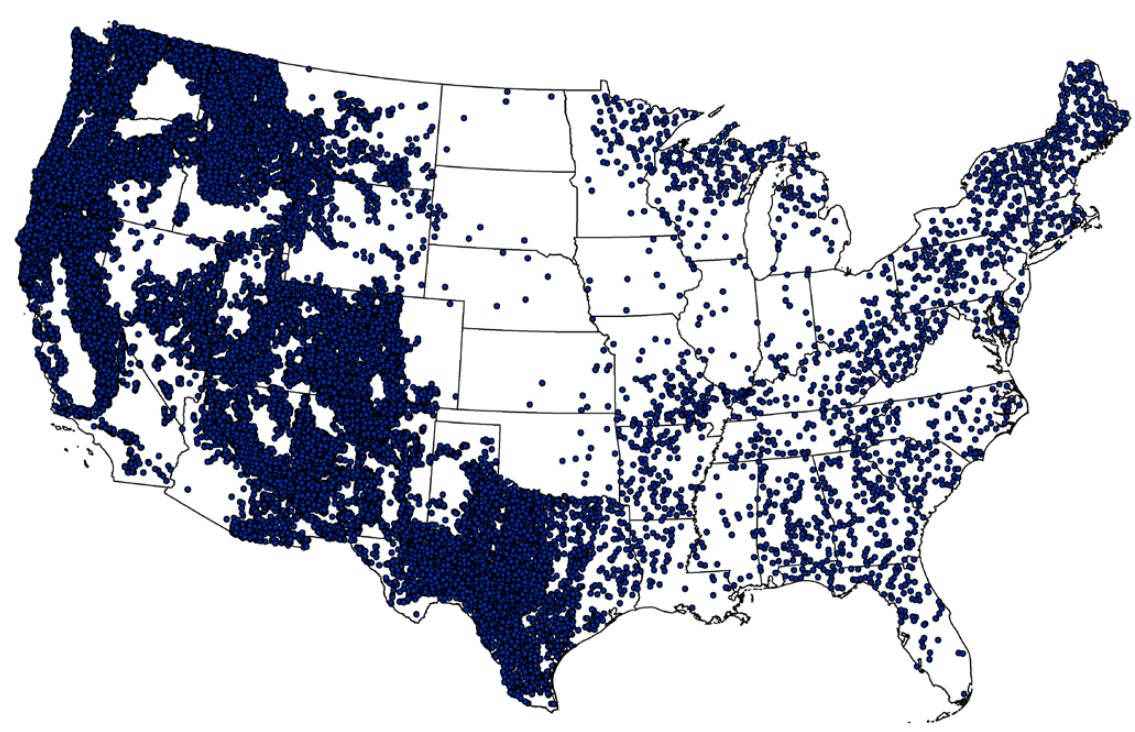

Figure 2. Location of FIA plots used to develop surface fuel loading classifications.





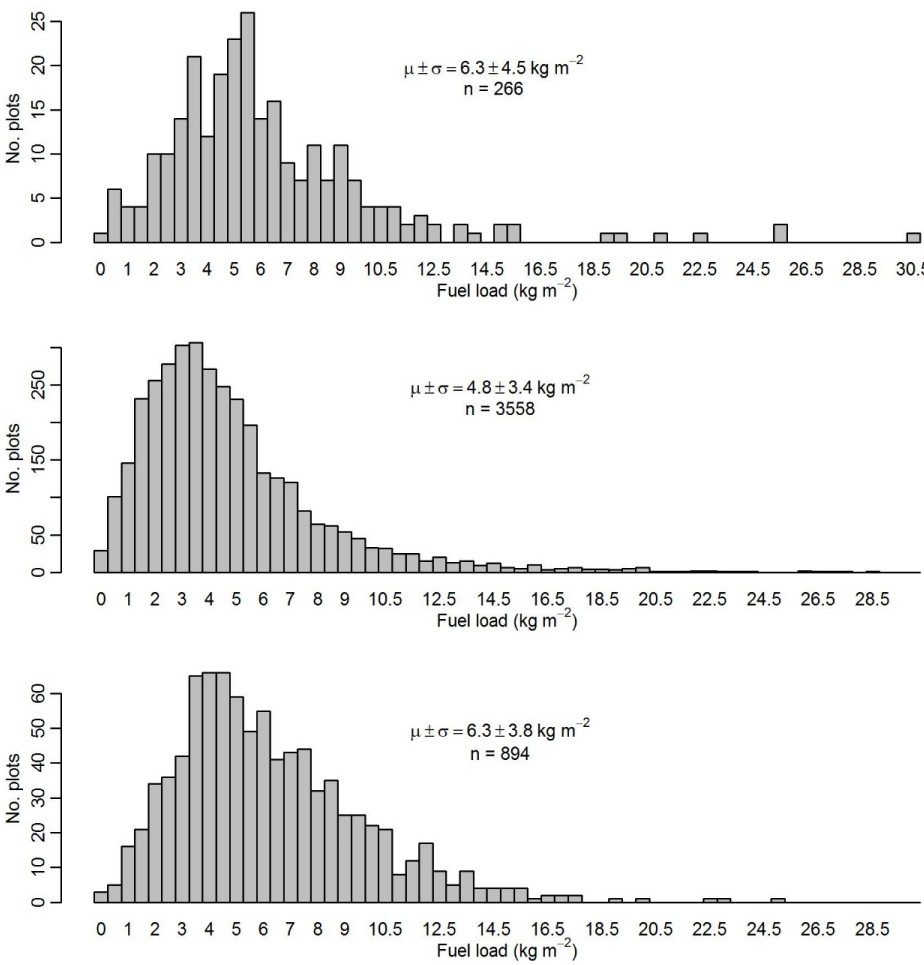

Figure 3. The distribution of surface fuel loading for the FIA plots of three FTG, Loblolly/shortleaf pine (160), Douglas-fir (200), and California mixed conifer (370).





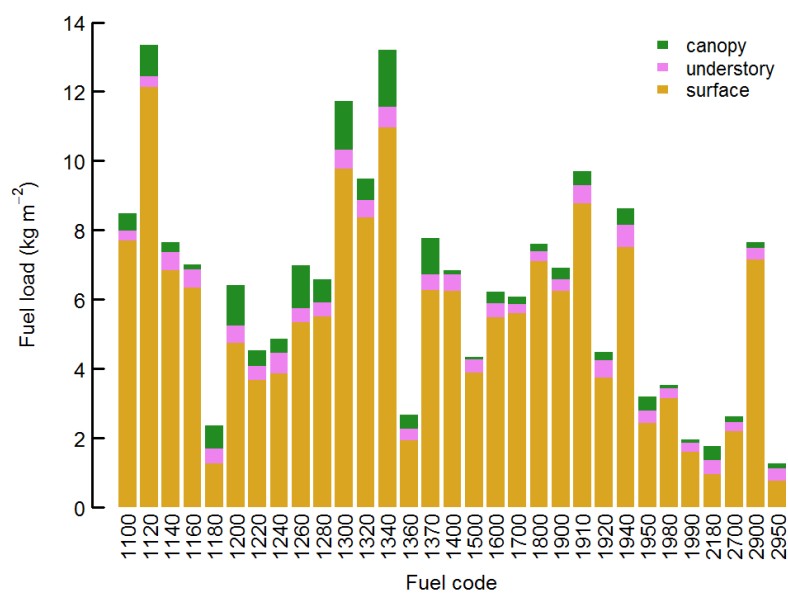

Figure 4. Best estimate (Table 4) forest fuel loading in canopy, understory, and surface fuels by fuel type.

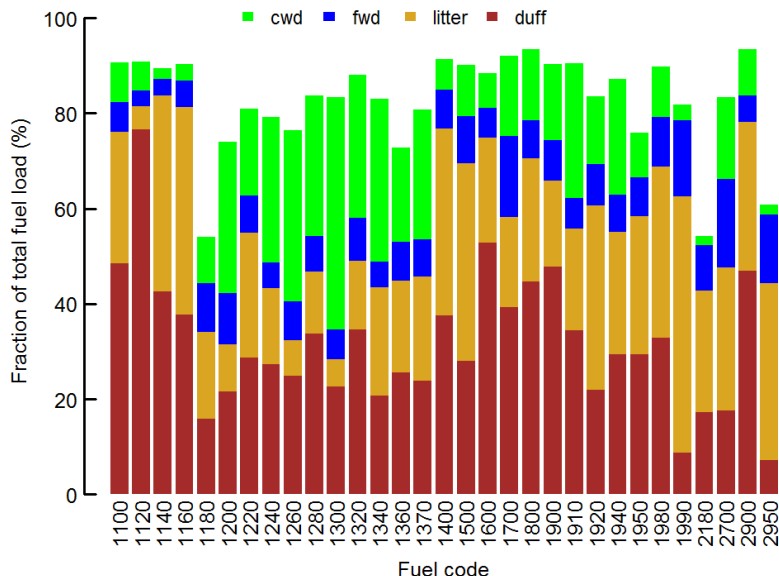

Figure 5. Fraction of best estimate total forest fuel loading in surface fuel loading groups (Table 4) by fuel type.



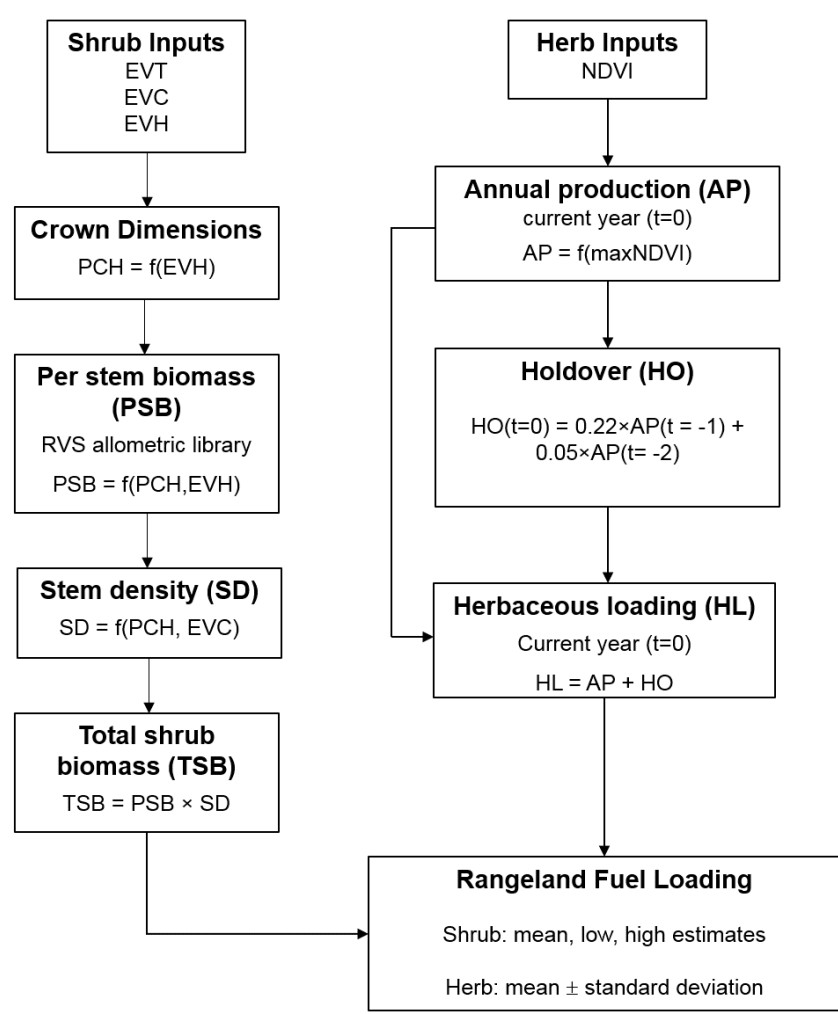

5  Figure 6. Abbreviated flow of data and actions in RVS to produce rangeland fuel loadings. EVT, EVC and EVH are Existing Vegetation Type, Existing Vegetation Cover, and Existing Vegetation Height from the Landfire Project.




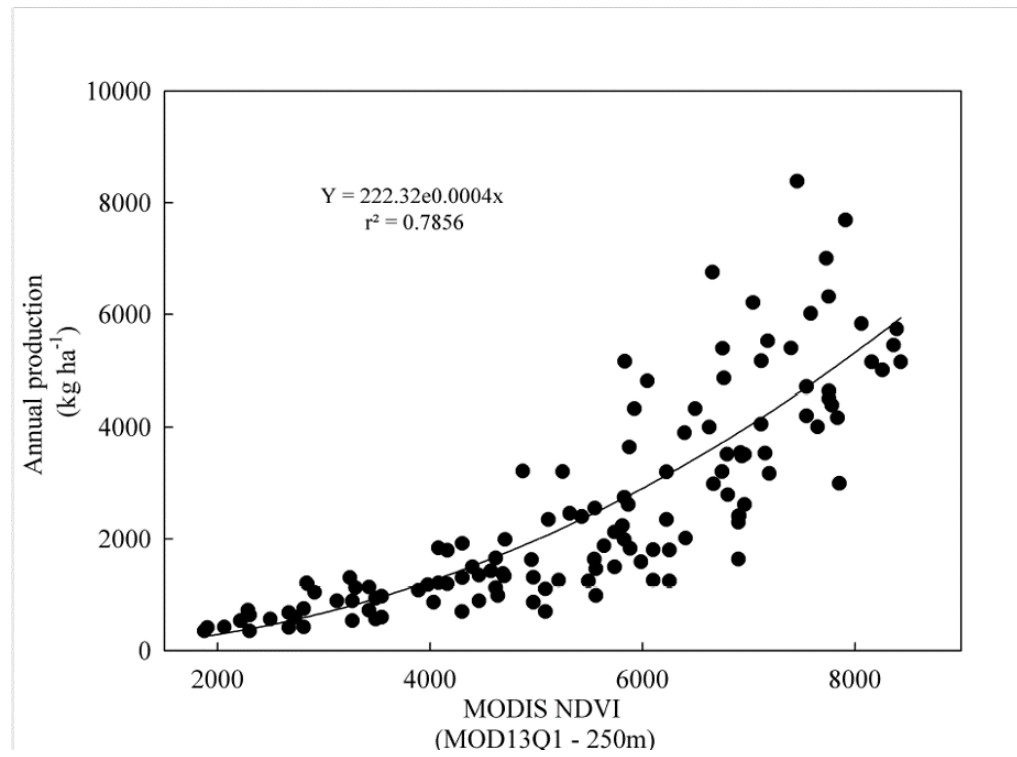

Figure 7. Relationship between annual production and annual maximum NDVI on 51 grassland vegetation types.



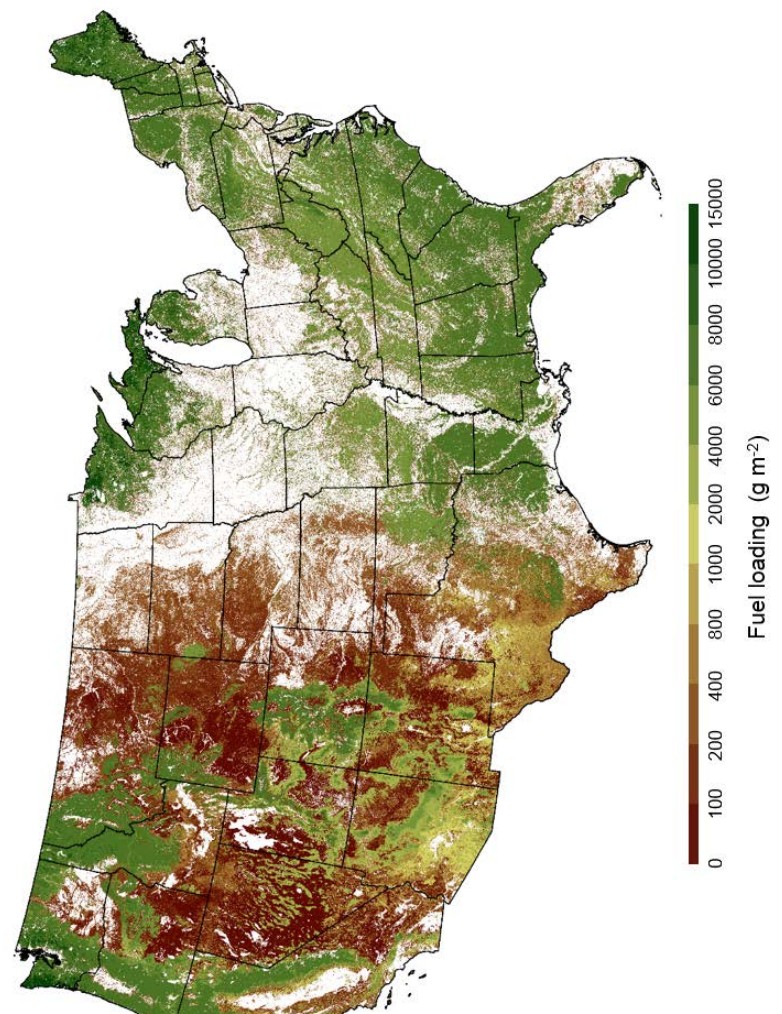

Figure 8. Map of best estimate fuel loading for forest and rangelands in g m$^{-2}$.





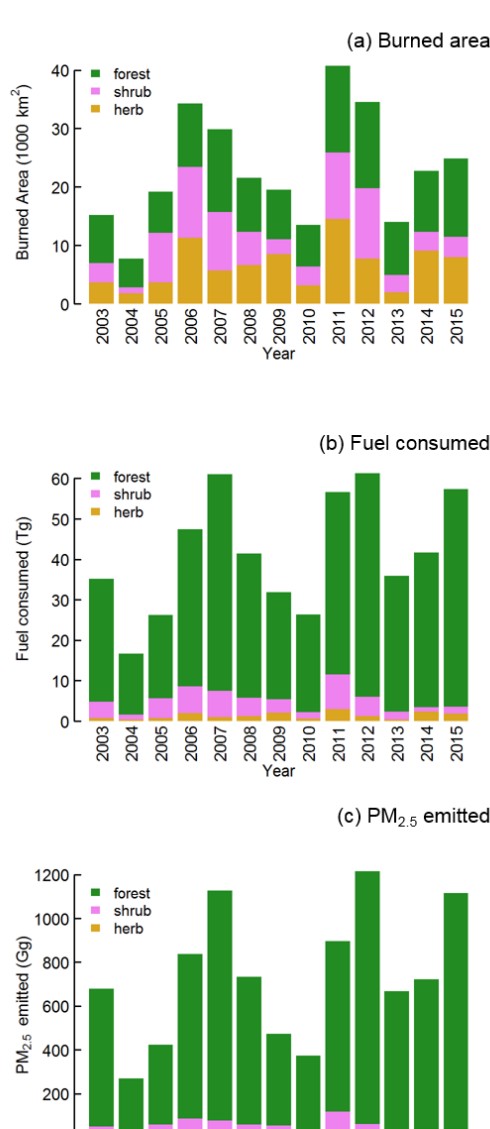

Figure 9. Annual burned area, fuel consumed, and PM$_{2.5}$ emitted for 2003-2015.





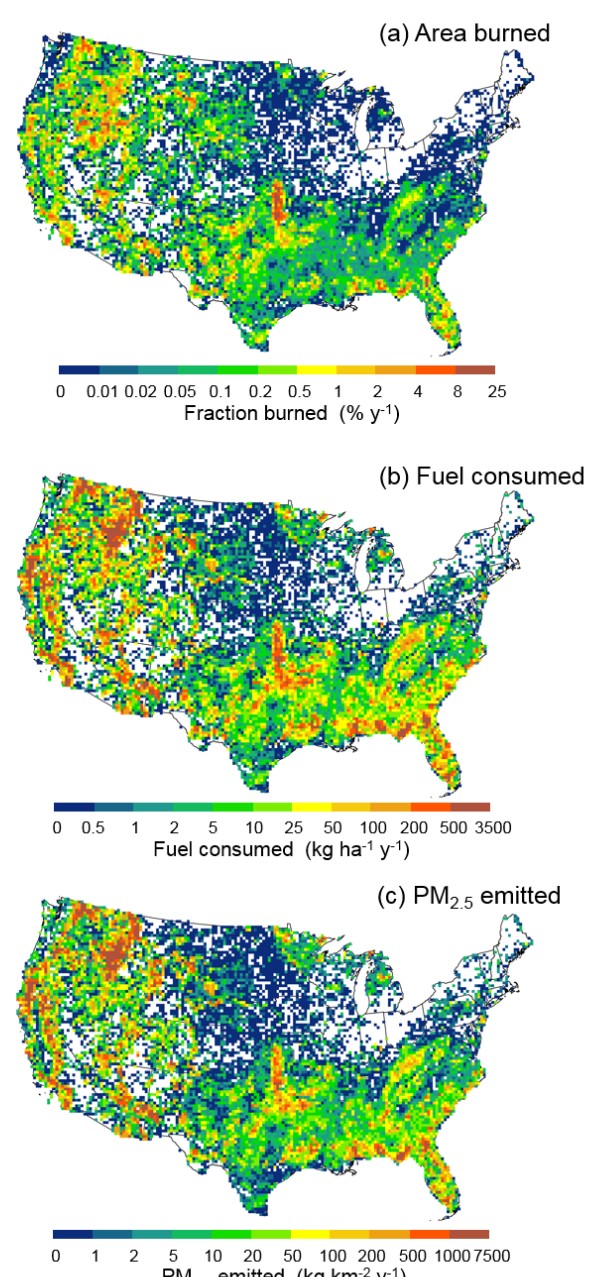

Figure 10. Annual burned area, fuel consumed, and $PM_{2.5}$ emitted averaged over 2003-2015.





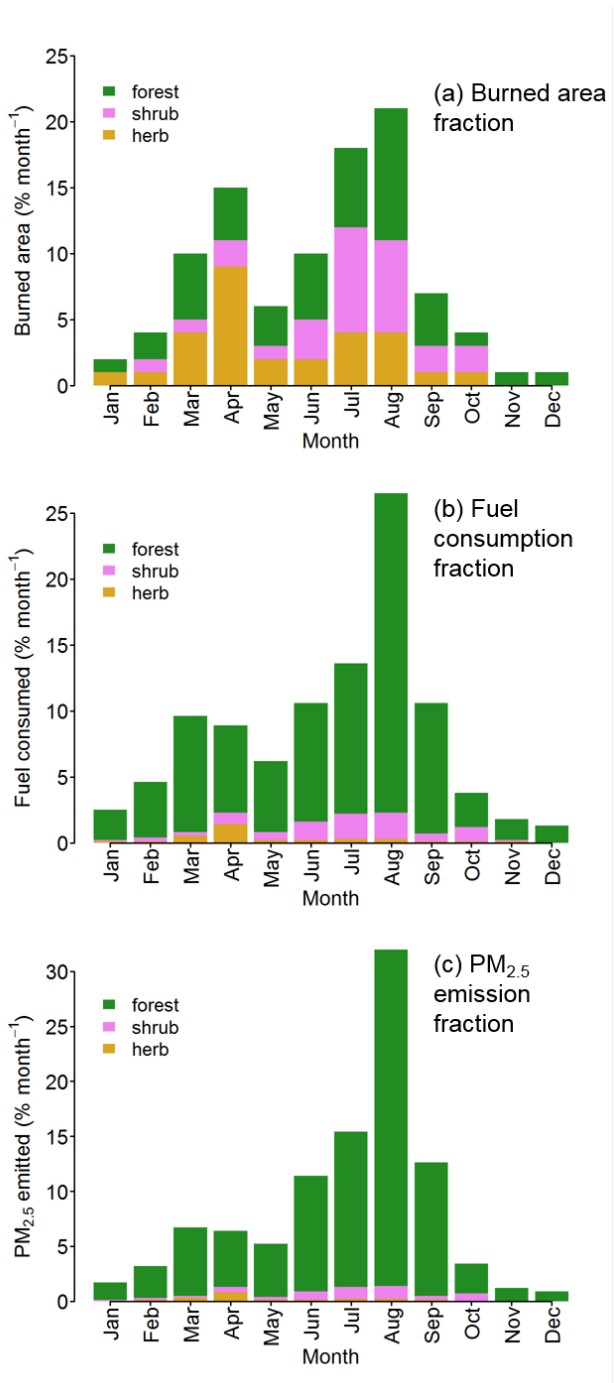

Figure 11. Monthly distributions of burned area, fuel consumption, and PM$_{2.5}$ emitted over 2003-2015, broken down by cover type.



Figure 12. Seasonal PM$_{2.5}$ emitted average over 2003-2015.



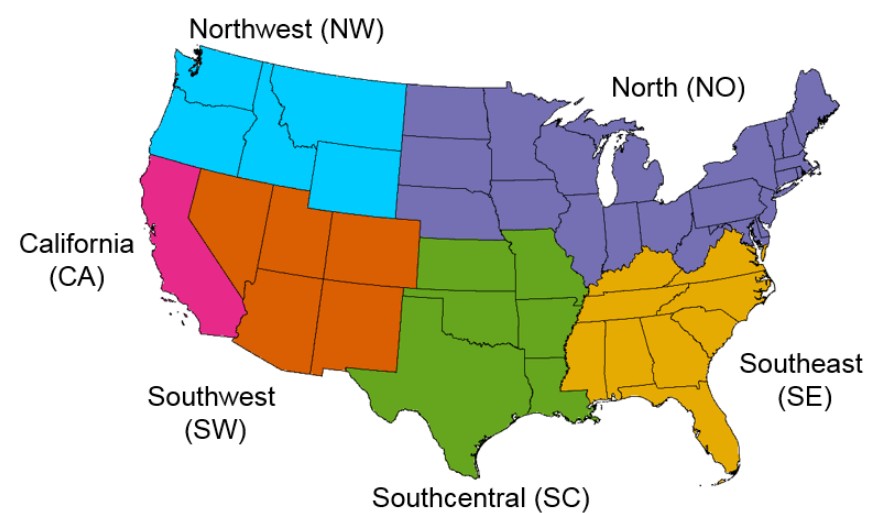

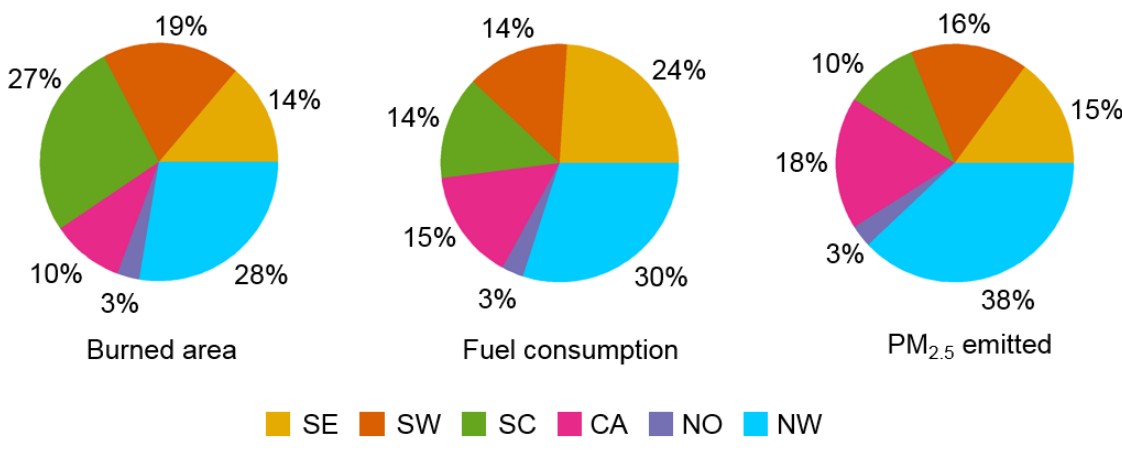

Figure 13. Top panel: geographic regions. Bottom panel: Burned area, fuel consumption, and PM$_{2.5}$ emitted by region.



Figure 14. Monthly $PM_{2.5}$ emitted averaged over 2003-2015.





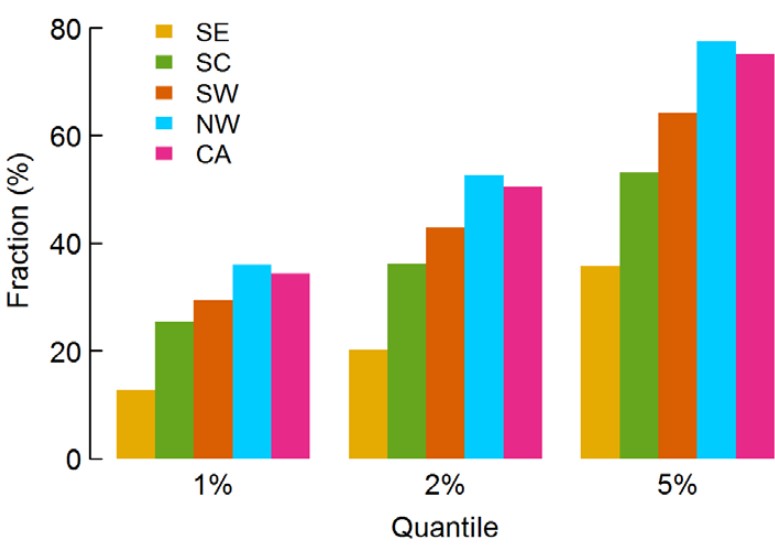

Figure 15. Fraction of regional, 2003-2015 PM$_{2.5}$ emissions released on peak days.




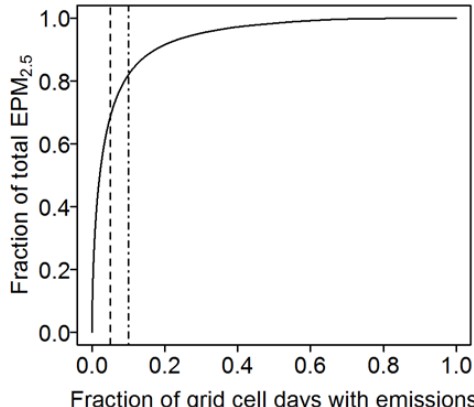

Figure 16. Cumulative distribution of daily PM$_{2.5}$ emissions aggregated on a 10 km × 10 km grid. Dashed line and dashed –
5    dotted line mark 5 % and 10 % of grid cell days with emissions.





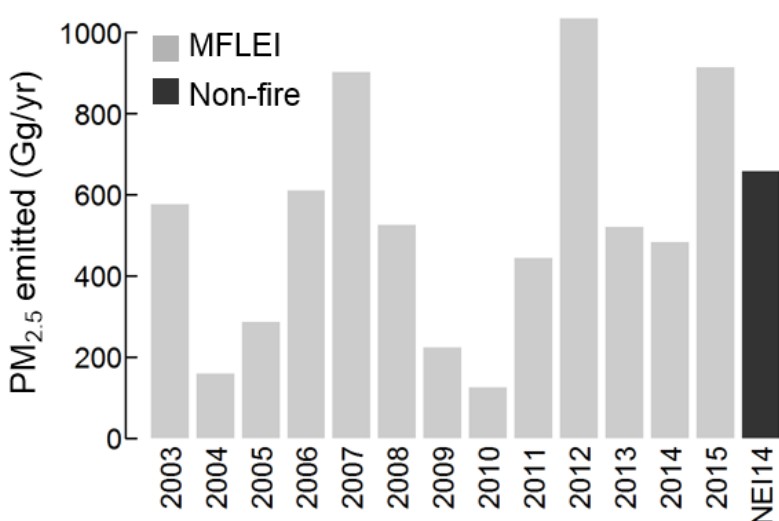

5    Figure 17. Annual PM$_{2.5}$ emitted in west.

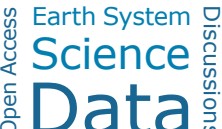

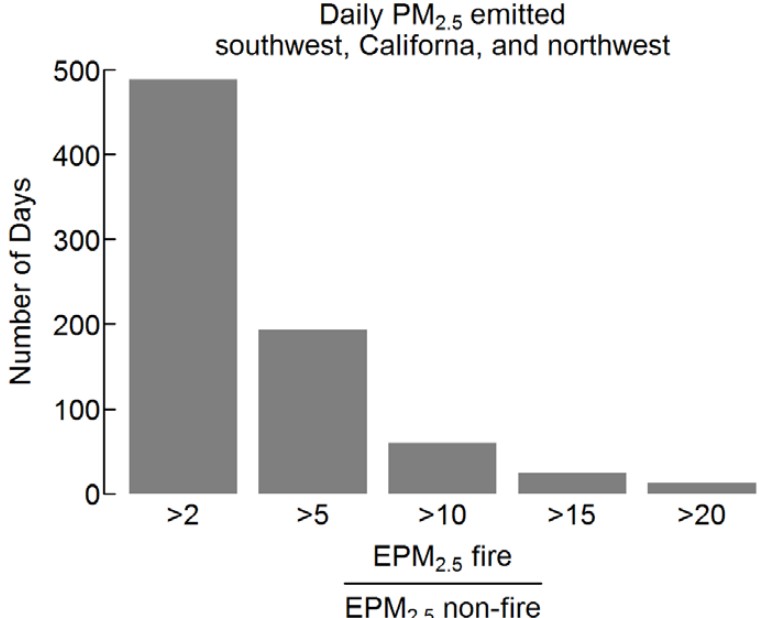

Figure 18. Number of days over 2003-2015 when the wildfire to non-wildfire PM$_{2.5}$ emission ratio in the west exceeds thresholds
5  of 2, 5, 10, 15, and 20.



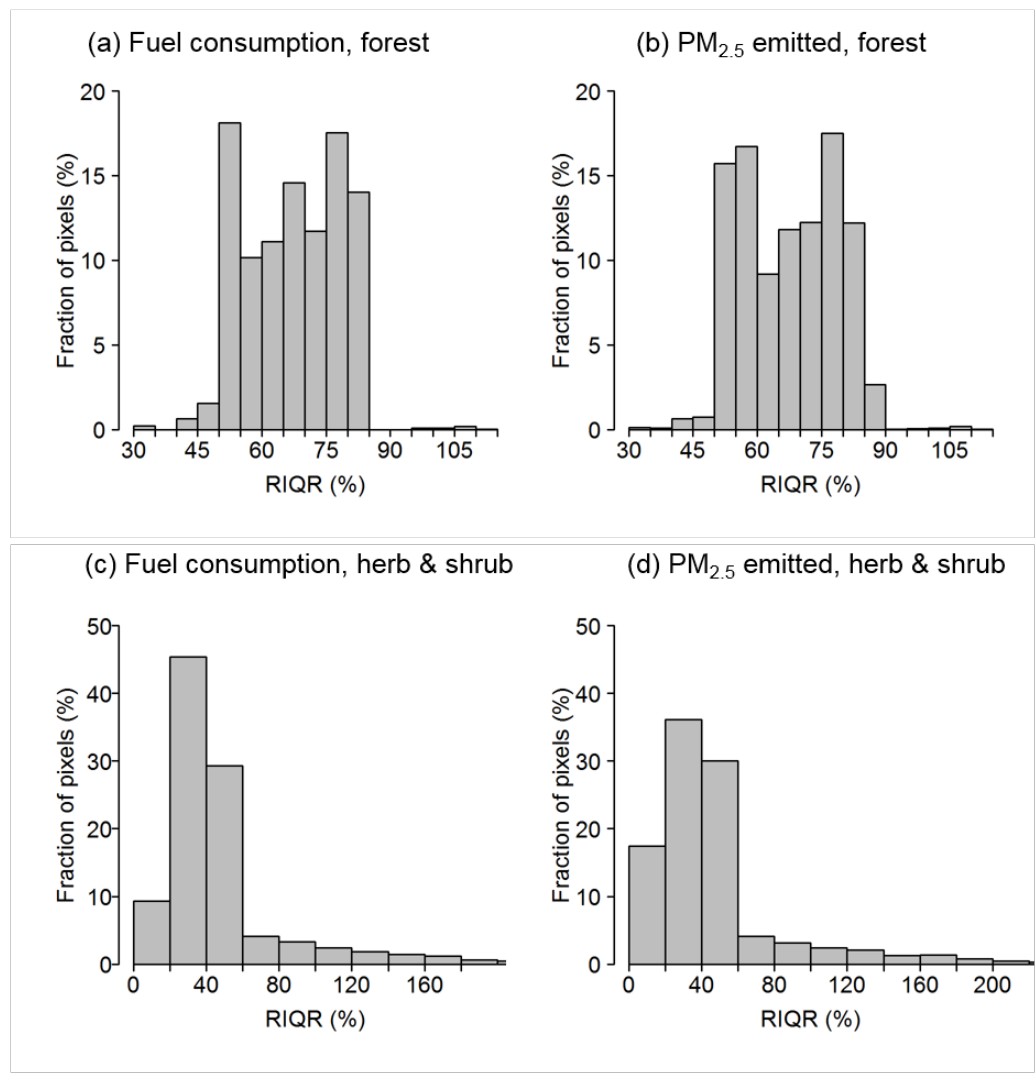

5    Figure 19. Distribution of relative interquartile range from pixel level Monte Carlo style simulations.



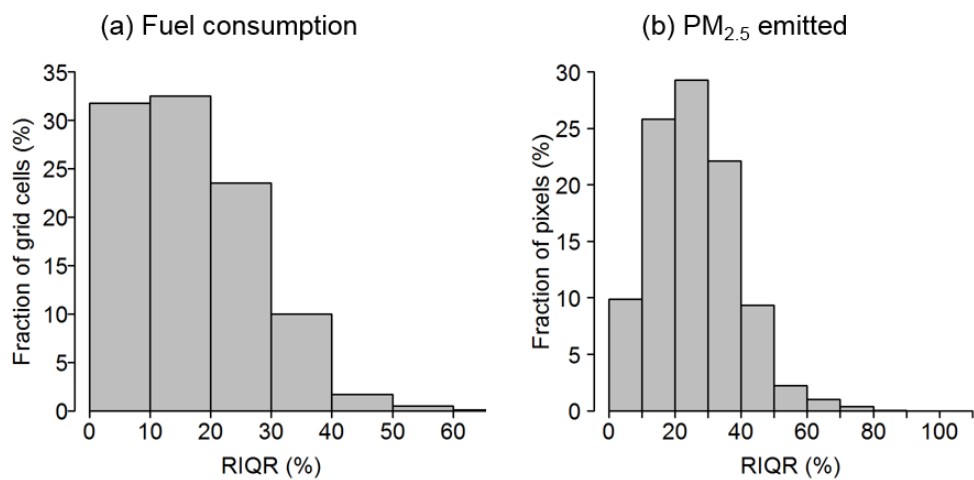

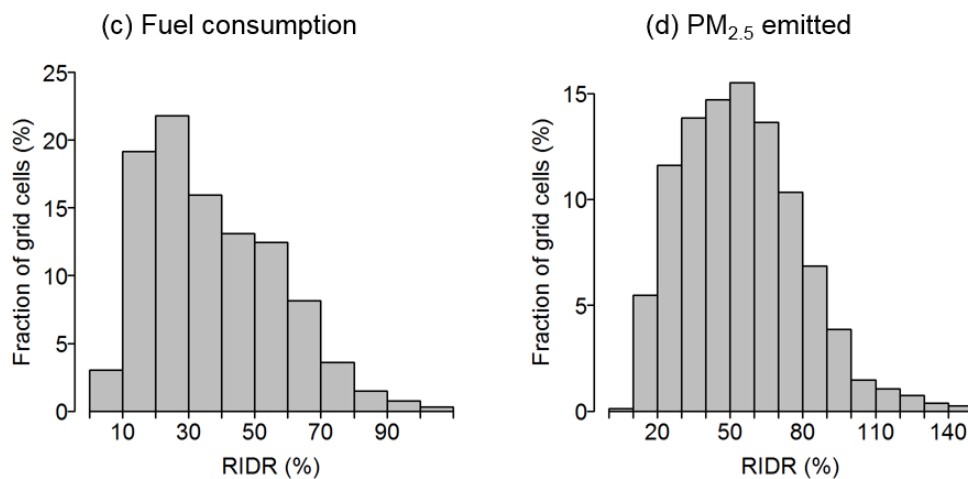

Figure 20. Distribution of relative interquartile range (top panel) and relative interdecile range (bottom panels) from 10 km × 10
5   km gridded Monte Carlo style simulations.





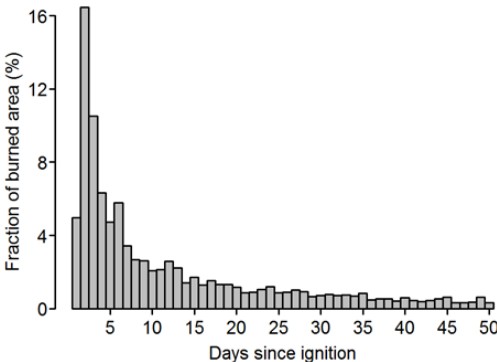

Figure A1. The burn day distribution for the 12,500–25,000 ha size class. Distributions for all six size classes are provided in the
dataset supplement (file\Supplements\BurnDayDist.csv).

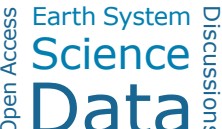

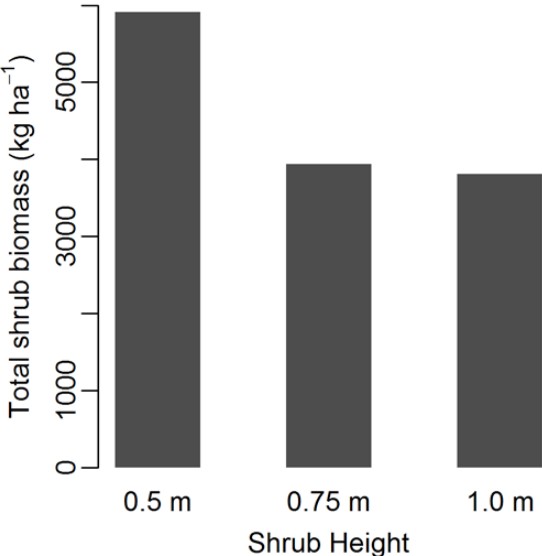

5    Figure C1. Total shrub biomass estimates for a pixel with EVT class of Big Sagebrush shrubland, EVH class of 105, and EVC class of 112 (see text).