# Peer review of "Contiguous United States wildland fire emission estimates during 2003-2015"

_Earth System Science Data, 2018_

## Referee Comment (RC1) · Anonymous Referee #1 · 7 Oct 2018

General Comments

The manuscript describes calculating the emissions for the contiguous US (CONUS) for 2003-2015. The authors rigorously describe a methodology and present the calculation

Having read through this manuscript, I am satisfied that it is complete and thorough and from my perspective, is acceptable in its current form. With that said, it is a very dense paper with a lot of data management and handling. Some discussions I fully grasped and were confident they used appropriate methods (e.g. fire areas and dates), while others sections (e.g. fuel loads) appear adequately handled but I cannot say for sure.

Specific Comments

[Figure]

In the introduction, the authors describe the various emissions inventories for the US (page 1, third paragraph). They may wish to mention

Larkin, N. K., Raffuse, S. M., & Strand, T. M. (2014). Wildland fire emissions, carbon, and climate: US emissions inventories. Forest Ecology and Management, 317, 61-69.

Also, the authors may wish to mention the work by Canadians, which follows a similar methodology to that presented in this manuscript

De Groot, W.J., Landry, R., Kurz, W.A., Anderson, K.R., Englefield, P., Fraser, R.H., Hall, R.J., Banfield, E., Raymond, D.A., Decker, V. and Lynham, T.J., 2007. Estimating direct carbon emissions from Canadian wildland fires1. International Journal of Wildland Fire, 16(5), pp.593-606.

Anderson, K., Simpson, B., Hall, R.J., Englefield, P., Gartrell, M. and Metsaranta, J.M., 2015. Integrating forest fuels and land cover data for improved estimation of fuel consumption and carbon emissions from boreal fires. International Journal of Wildland Fire, 24(5), pp.665-679.

On page 2, line 26, when the authors state "each burned grid cell is burned in its entirety", I assume the authors are referring to spatial extent (ha) and not fuel load (tonnes).

Under 2.2 Land cover, are there not several US land cover maps (NFDRS, Hardy, LANDFIRE, Ok-Wen, FCCS), that produce different fuel loads? The authors may wish to reference these and justify their choice.

Under 2.3.3 Unburned and lightly burned grid cells, the authors describe the 6 BSEV categories inside the fire polygon. I am not clear on how a category of increasing green would be mapped inside a fire polygon. Presumably this would have described in the referenced paper (Eidenshink et al., 2007) but it would be helpful to briefly describe the process (perhaps in 2.3.1).

Technical Corrections None

---

## Referee Comment (RC2) · Anonymous Referee #2 · 14 Oct 2018

General comments

Large wildfires have increased dramatically in many western U.S. regions under the century droughts that have lasted for nearly two decades, emitting a large amount of air pollutants and impacting air quality and human health not only at the burned sites and surrounding areas but also in remote downwind regions. There are urgent needs to simulate and evaluate these impacts according to the EPA NAAQS which includes standards for PM2.5 and O3 at daily scale. The high-resolution daily wildfire emission inventory in CONUS developed in this study, the Missoula Fire Lab Emission Inventory (MFLEI), is very valuable for the smoke modeling and impact assessment efforts. The result analyses provided in this study are useful for improving our understanding of the features of fuel loading, fire consumption, and emissions in CONUS.

Specific comments

1. A large number of fuel, fire, and other sources are used when estimating fire emissions based on Eq.1. It would be helpful to provide a diagram to summary the major sources and connections.

2. Comparisons are provided between this inventory and several previous ones in the introduction section. It would be useful to briefly compare the results, especially with the previous daily inventory.

3. This new inventory provides daily emissions. Surface fuels at 10- and 1-hr vary at this scale. Why fuel moistures of 1000-h and 100-h rather than 10- and 1-hr fuels are used?

4. This inventory provides 250-m fire emissions. Fuel moisture is obtained from NFDRS station. What is the resolution of the NFDRS station and how could the resolution mismatch between the fire emission and NFDRS station affect the emission estimates?

5. It is indicated that MFLEI will be updated, with recent years, as the MTBS burned area product becomes available. MFLEI also uses other fire sources such as FOD. What would be the impacts if FOD is not updated in the future?

6. Subsection 3.5: The title includes "agricultural fires" but they are not discussed in this subsection.

7. Section 5: It is more like a summary than conclusions.

---

## Author Comment (AC1) · 19 Nov 2018

Referee #1 Comments

RCX. Denotes the referee comment

ARX. Denotes the authors' response including associated manuscript revisions

RC1. In the introduction, the authors describe the various emissions inventories for the US (page 1, third paragraph). They may wish to mention Larkin, N. K., Raffuse, S. M., & Strand, T. M. (2014). Wildland fire emissions, carbon, and climate: US emissions inventories. Forest Ecology and Management, 317, 61-69.

Also, the authors may wish to mention the work by Canadians, which follows a similar

methodology to that presented in this manuscript

De Groot, W.J., Landry, R., Kurz, W.A., Anderson, K.R., Englefield, P., Fraser, R.H., Hall, R.J., Banfield, E., Raymond, D.A., Decker, V. and Lynham, T.J., 2007. Estimating direct carbon emissions from Canadian wildland fires1. International Journal of Wildland Fire, 16(5), pp.593-606.

Anderson, K., Simpson, B., Hall, R.J., Englefield, P., Gartrell, M. and Metsaranta, J.M., 2015. Integrating forest fuels and land cover data for improved estimation of fuel consumption and carbon emissions from boreal fires. International Journal of Wildland Fire, 24(5), pp.665-679.

AR1. We have added the Larkin et al. reference to P3, line 31. The revised text reads:

"Several biomass burning emission inventories that include CONUS are available (van der Werf et al., 2017; Zhang et al., 2017; French et al., 2014; Larkin et al., 2014; Wiedinmyer et al., 2011)."

We have also referenced the Canadian wildfire emission inventories at Page 4, Line 1. The text now reads:

"MFLEI uses a forest type map and a new forest fuel classification, both of which are based on a national forest inventory dataset, providing more accurate fuel loading estimates compared to the fuels layer used in WFEIS (Keane et al., 2013). The methodology used to develop MFLEI is similar to that employed to develop carbon emission estimates for Canadian wildland fires (Anderson et al., 2015; De Groot et al., 2007). As a retrospective inventory, MFLEI is able to leverage geospatial fire activity information including high spatial resolution burned area and burn severity products that are not available for real-time inventories (e.g. FiNN)."

RC2.On page 2, line 26, when the authors state "each burned grid cell is burned in its entirety", I assume the authors are referring to spatial extent (ha) and not fuel load (tonnes).

[Figure]

AR2. The reviewer is correct. We did not intend to imply that all fuel present was burned. The text has been changed to: "The inventory assumes that the burning and emissions for each burned grid cell occur on the estimated burn day (Sect. 2.3.2)."

RC3. Under 2.2 Land cover, are there not several US land cover maps (NFDRS, Hardy, LANDFIRE, Ok-Wen, FCCS), that produce different fuel loads? The authors may wish to reference these and justify their choice.

AR3. The reviewer is correct, there are several CONUS wide maps of land cover and fuel type. The LANDFIRE Project (https://www.landfire.gov/data_overviews.php) has created many geospatial data products including fire behavior fuel models (FBFM), which include the model used for NFDRS, vegetation type, and surface fuel loading models (FCCS and FLM). We assembled our own land cover map so we could use the large dataset (>27,000 plots) of USFS Forest Inventory and Analysis Program vegetation and fuels data for forests and use fuel loading from the Rangeland Vegetation Simulator (RVS) for grasslands and shrublands. The RVS map and fuel loading was developed using LANDFIRE products along with MODIS NDVI and rangeland productivity data as described in Sect. 2.4.2. Our justification for assembling our own land cover map is detailed in following text which has been added to Section 2.2 of the manuscript on Page 5, Line 4:

"The LANDFIRE project (LANDFIRE, 2016) provides CONUS wide maps for Fuel Characteristics Classification System (FCCS; Ottmar et al., 2007; McKenzie et al., 2012) and Fuel Loading Models (FLM, Lutes et al., 2006) fuelbed models, both of which are suitable for estimating fuel consumption and emissions. FCCS is used in both the NEI+ (Larkin et al., 2014) and WFEIS (French et al., 2014) CONUS fire emission inventories. We assembled a new map based on the USFS forest type group map because it provides three important benefits over other land cover maps with respect to forests. First, the accuracy of the forest type group map is significantly better than either the FCCS or FLM maps (Keane et al., 2013). Second, it enabled us to use the Fuels Type Group (FTG) surface fuel classification system (Sect. 2.4.1) which provides a more accurate

estimate of average surface fuel loading than either the FCCS or FLM (Keane et al., 2013). Finally, because the USFS forest type group classification is an FIA plot variable, we are able to use the large (>27,000 plots) dataset of FIA fuel measurements estimate uncertainty in surface fuel loading and emissions (Sect. 2.9)."

RC4. Under 2.3.3 Unburned and lightly burned grid cells, the authors describe the 6 BSEV categories inside the fire polygon. I am not clear on how a category of increasing green would be mapped inside a fire polygon. Presumably this would have described in the referenced paper (Eidenshink et al., 2007) but it would be helpful to briefly describe the process (perhaps in 2.3.1).

AR4. The burn severity classification of increased greenness is very rare. During our period (2003-2015) only 0.3% of MTBS pixels were classified as increased greenness. Given the rare occurrence of the increased greenness classification, it has negligible effect on our emission product. The MTBS burn severity class data are derived from Landsat imagery by analysis of a pre-fire scene and a post-fire scene to create a Differenced Normalized Burn Ratio (dNBR) image (as described in Eidenshink et al., 2007). For some fires, an increased response in vegetation productivity, results in increased greenness. This could results from an area that did not burn and was greener at the time of the post-fire scene than it was pre-fire scene. It is not uncommon for the pre-fire scene to be from the previous year. In which case an area that did not burn or was very lightly burned may have increased greenness compared to the previous year due to increased productivity or other factors. The availability of optimal Landsat scenes is limited by the 16-day Landsat revisit cycle, atmospheric conditions (clouds, smoke from active fires, terrain shadows), and factors such as sun angle and length of growing season limit the availability of optimal scenes for analysis (https://www.mtbs.gov/mapping-methods).

Given the rare occurrence (0.3% of pixels) and negligible effect of the increased greenness classification, we believe that an explanation is not warranted in the text. We have revised the text clarifying that the increased greenness classification is very rare.

In Sect. 2.3.3 Unburned and lightly burned pixels, Page 8, line 14, following the sentence "We elected to designate BSEV = 1 as unburned, which is consistent with MTBS program publications that describe this classification as areas which are either unburned or where visible fire effects occupy < 5 % of the site at the time of observation (Schwind, 2008)." we have added the text:

"The increased green classification may indicate unburned that exhibited more green at the time of the post-fire Landsat scene relative to the pre-fire scene. The increased green classification was assigned to just 0.3% of MTBS pixels and thus has a negligible impact on our inventory."

Lutes, D. C., Keane, R. E. and Caratti, J. F.: A surface fuel classification for estimating fire effects, Int. J. Wildland Fire, 18(7), 802–814, doi:10.1071/WF08062, 2009. McKenzie, D., French, N. H. F. and Ottmar, R. D.: National database for calculating fuel available to wildfires, Eos, Transactions American Geophysical Union, 93(6), 57–58, doi:10.1029/2012EO060002, 2012.

---

## Author Comment (AC2) · 19 Nov 2018

Referee #2 Comments and Authors' Responses

Specific comments

RC1. A large number of fuel, fire, and other sources are used when estimating fire emissions based on Eq.1. It would be helpful to provide a diagram to summary the major sources and connections.

AR1. We have added a diagram which summarizes the main steps of the inventory methodology and highlights the connections of the multiple datasets to the process. The diagram has been added as Figure 1. The text in Sect. 2.1 has been revised (Page 4, Line 14) with the insertion of the following sentence:

[Figure]

"The MFLEI biomass burning emission model is based on Eq. (1), given below, and the implementation and datasets are summarized in Figure 1."

Figure 1 has been included as jpg attachment with this response

RC2. Comparisons are provided between this inventory and several previous ones in the introduction section. It would be useful to briefly compare the results, especially with the previous daily inventory.

AR2. We have added a section comparing MFLEI with three other emission inventories that are mentioned in the introduction section: GFED, FINN, and WFEIS. The revised text is given below. Two figures and two tables have been added as part this revision and have been attached as jpg and pdf.

3.6 Comparison with other emission inventories Next we compare the estimated fuel consumption and PM2.5 emissions of MFLEI with three fire emissions inventories: GFED v4.1s (GFED, 2018), FINN v1.5 (FINN, 2018), and WFEIS v0.5 (WFEIS, 2018). In this comparison we have excluded fuel consumption and PM2.5 emissions associated with agricultural burning from all three inventories. Regional annual fuel consumption from the four inventories is plotted in Figure 21. Statistics comparing MFLEI regional annual fuel consumption versus the other inventories are given in Table 11. There is significant variability in the agreement between MFLEI and the other inventories. Across the west (NW, CA, SW), MFLEI annual fuel consumption is well correlated with both FINN and GFED (Table 11). MFLEI fuel consumption exceeds the mean of FINN, GFED, and WFEIS in nearly all years and is generally the highest in Northwest and Southwest regions (Fig. 21a). In the east regions (SC, SE, NO), MFLEI fuel consumption fluctuates about the FINN/GFED/WFEIS mean value (Fig. 21b). In terms of variability and mean absolute relative difference, MFLEI agrees best with GFED. Regional annual PM2.5 emissions are shown in Figure 22 and statistics comparing MFLEI PM2.5 emissions versus the other inventories are given in Table 12. As with fuel consumption, across the west (NW, CA, SW), MFLEI annual PM2.5 emissions

are well correlated with both FINN and GFED, while correlation with WFEIS is weak in most regions (Table 12). In the west, MFLEI annual PM2.5 emissions are highest among the inventories in most years (Fig. 22a). The greater PM2.5 emissions of MFLEI in the west are partly attributable to the use of a larger EFPM2.5 for western forests (22.8 g kg-1, Table 9) compared with FINN (12.9 g kg-1), GFED (12.6 g kg-1), and WFEIS (11.9 g kg-1). (Because WFEIS uses combustion phase dependent EFs applied in a non-transparent manner, we have taken EFPM2.5 as the ratio of the sum of EPM2.5 to the sum of fuel consumed for all western forests.) MFLEI uses EFPM2.5 from the synthesis of Urbanski (2014) that accounts for the lower MCE measured for wildfires in western conifer forests (Urbanski, 2013). FINN and GFED use EFPM2.5 from Akagi et al (2011), with updates from May et al. (2014), which are based on emission measurements of prescribed fires, most of which occurred in the Southeast US. WFEIS employs EFPM2.5 measured for prescribed burns of logging slash. The higher EFPM2.5 used by MFLEI for wildfires in western forests is consistent with recent emission measurements of Lui et al. (2017). In a study of western US wildfires, Lui et al. (2017) reported an average EFPM1 = 26.0 g kg-1 (PM1 = particulate matter with an aerodynamic diameter < 1 $\mu$m), more than 2 times the EF for prescribed fires.

RC3. This new inventory provides daily emissions. Surface fuels at 10- and 1-hr vary at this scale. Why fuel moistures of 1000-h and 100-h rather than 10- and 1-hr fuels are used?

AC3. We estimated fuel consumption of grass, shrubs, and down dead wood using the natural fuel algorithms from the CONSUME model. These CONSUME algorithms simulate consumption completeness independent of fuel moisture for grass, shrubs, and down dead wood in the 1-h (< 1 cm diameter), 10-h (1-2.5 cm diameter), and 100-h (2.5-7.6 cm diameter) size classes. The CONSUME algorithms do use 1000-h fuel moisture and duff moisture for simulating combustion completeness for down dead wood in the 1000-h size class. Combustion completeness for litter was based on the FOFEM model, which for wildfires estimates litter consumption independent of

moisture content. We used the 100-h fuel moisture to estimate duff moisture based on Harrington (1982) (Page 13, L27 of manuscript). The duff moisture estimated from 100-h fuel moisture was used in the FOFEM duff consumption equations and in the CONSUME down dead wood equations that used duff moisture as a variable. The 1-h and 10-h fuel moistures are very important for estimating/simulating fire spread rates since fuels in these size classes, grasses, litter, and fine woody debris, are key drivers of fire spread (Albini 1976; Rothermel, 1972). Since MFLEI is a retrospective emission inventory we do not need to predict fire spread and therefore 1-h and 10-h are not used.

RC4. This inventory provides 250-m fire emissions. Fuel moisture is obtained from NFDRS station. What is the resolution of the NFDRS station and how could the resolution mismatch between the fire emission and NFDRS station affect the emission estimates?

AR4. The NFDRS stations are irregularly spaced (for current locations see https://www.wfas.net/index.php/fire-weather-stations-static-maps-43) and some stations operate/report data only during the station's regional fire season. The median distance between nearest NFDRS stations was ïA¿28 km.

If the fuel moisture regime was in error by one category (e.g. fuel consumption was modeled using 1000-h and duff moisture of "dry" regime, but actual conditions were "moist" regime) the error in total fuel consumption would range between +/- 2% and +/- 12%, depending on the forest type and direction of error in fuel moisture regime. For all years of the inventory, if the fuel moisture regime used was systemically one category lower (drier) than the actual moisture regime for all burned forest pixels, the overestimate in total forest fuel consumption would be ïA¿5%. Emission are directly proportional to fuel consumption.

RC5. It is indicated that MFLEI will be updated, with recent years, as the MTBS burned area product becomes available. MFLEI also uses other fire sources such as FOD. What would be the impacts if FOD is not updated in the future?

AR5. Dr. Karen Short, creator of FOD will be releasing an update with 2016 and 2017 at the end of this year (2018). If FOD is not updated beyond 2017, there would be a minor impact on MFLEI. We used FOD to include burned area from wildfires not captured by MTBS, GEOMAC, and MCD64. Over 2003-2015, 8% of total MFLEI burned area was attributable to FOD. In the future, if FOD is unavailable MFLEI would miss roughly 10% of wildfire burned area. MFLEI also used FOD to assign containment dates to MTBS fires and discovery dates to GEOMAC fires (recall MCD64 product provides the estimated day of burning for each pixel). Fortunately, discovery dates and containment dates are available for most MTBS and GEOMAC fires from one of five national databases (USDI Wildland Fire Management Information System, FWS Fire Management Information System, USFS Fire Statistics, USFA National Fire Incident Reporting System, and National Association of State Foresters). (In FOD, the information for ïA¿80% of all CONUS wildfires >10 acres was obtained from one of these five national databases (Short, 2014; Short, 2017)). If FOD is unavailable, we will extract much of the needed information from the five national fire databases listed above after consultation with Dr. Karen Short who developed FOD and is a USFS research colleague of the MFLEI team.

RC6. Subsection 3.5: The title includes "agricultural fires" but they are not discussed in this subsection. AR6. The title of subsection 3.5 has been changed to: "Prescribed fires" since agricultural fires are excluded from MFLEI and are not discussed in this section.

RC7. Section 5: It is more like a summary than conclusions.

AR7. We agree with the referee that Section 5 is largely a summary of the paper. However, we believe the content and tone is appropriate for a conclusion section of a dataset paper. We have reviewed the conclusion section of several papers published in ESSD and found ours to similar in content and tone, see for example e.g. Chuvieco et al., 2018, 10, 2015-2031. We have revised the Section 5 to mention the comparison of MFLEI with GFED, FINN, and WFEIS. The additional text is:

"A regional comparison of MFLEI with three fire emission inventories, FINN v1.5, GFED v4.1s, and WFEIS v0.5, showed MFLEI predicted significant greater PM2.5 emissions across the west, in part due to the use of a larger EFM2.5 for wildfires in forests."

References Albini, Frank A. 1976. Estimating wildfire behavior and effects. Gen. Tech. Rep. INT-GTR-30. Ogden, UT: U.S. Department of Agriculture, Forest Service, Intermountain Forest and Range Experiment Station. 92 p. Available: https://www.fs.usda.gov/treesearch/pubs/29574

Rothermel, Richard C. 1972. A mathematical model for predicting fire spread in wildland fuels. Res. Pap. INT-115. Ogden, UT: U.S. Department of Agriculture, Intermountain Forest and Range Experiment Station. 40 p. Available: https://www.fs.usda.gov/treesearch/pubs/32533

Short, Karen C. 2017. Spatial wildfire occurrence data for the United States, 1992-2015 [FPA_FOD_20170508]. 4th Edition. Fort Collins, CO: Forest Service Research Data Archive. https://doi.org/10.2737/RDS-2013-0009.4

Short, K. C. 2014. A spatial database of wildfires in the United States, 1992-2011. Earth System Science Data. 6: 1-27.

Please also note the supplement to this comment:
https://www.earth-syst-sci-data-discuss.net/essd-2018-100/essd-2018-100-AC2-supplement.pdf

[Figure]

**Fig. 1.**

[Figure]

[Figure]

[Figure]

**Fig. 2.**

[Figure]

**Fig. 3.**

[Figure]

**Fig. 4.**
* * *
**Interactive**
**comment**

[Figure]

**Fig. 5.**

**Supplement:**

Table 11. Statistics for comparison of annual fuel consumption by region between MFLEI and FINN v1.5, GFED v4.1s, and WFEIS v0.5. Regions are as defined in Fig. 14a.

| | Region | | | | | | |
|---|---|---|---|---|---|---|---|
| | CONUS | NW | CA | SW | NO | SC | SE |
| MFLEI versus FINN v1.5 (2003–2015) | | | | | | | |
| Mean | | | | | | | |
| RD[a] | -17% | 6% | 50% | 103% | -35% | -65% | -75% |
| Min RD | -71% | -94% | -25% | 61% | -103% | -131% | -135% |
| Max RD | 41% | 81% | 115% | 131% | 68% | 21% | -31% |
| r[b] | 0.62 | 0.90 | 0.87 | 0.92 | 0.57 | 0.24 | 0.70 |
| MFLEI versus GFED 4.1s (2003–2015) | | | | | | | |
| Mean RD | 29% | 14% | 3% | 75% | 16% | 35% | 43% |
| Min RD | 0% | -4% | -27% | 41% | -83% | -45% | -1% |
| Max RD | 60% | 40% | 52% | 105% | 90% | 91% | 76% |
| r | 0.90 | 0.97 | 0.96 | 0.97 | 0.62 | 0.79 | 0.76 |
| MFLEI versus WFEIS v0.5 (2003–2013) | | | | | | | |
| Mean RD | -2% | 30% | -26% | 130% | -99% | -51% | 40% |
| Min RD | -41% | -110% | -177% | 35% | -161% | -175% | -104% |
| Max RD | 56% | 137% | 112% | 196% | -17% | 121% | 181% |
| r | 0.95 | 0.43 | -0.20 | 0.88 | 0.20 | -0.34 | 0.06 |

[a]

$$RD = 100 \times \frac{X(t)_{MFLEI} - Y(t)_i}{0.5 * (X(t)_{MFLEI} + Y(t)_i)}$$

$X(t)_{MFLEI}$ = MFLEI fuel consumed in year = t

$Y(t)_i$ = i fuel consumed in year = t, where i = FINN, GFED, or WFEIS

[b]r = correlation coefficient

Table 12. Statistics for comparison of annual PM$_{2.5}$ emitted consumption by region between MFLEI and FINN v1.5, GFED v4.1s, and WFEIS v0.5. Regions are as defined in Fig. 14a.

| | | | | Region | | | |
|---|---|---|---|---|---|---|---|
| | CONUS | NW | CA | SW | NO | SC | SE |
| **MFLEI versus FINN v1.5 (2003–2015)** | | | | | | | |
| Mean | | | | | | | |
| RD[a] | 98% | 56% | 85% | 136% | 24% | -55% | -70% |
| Min RD | -70% | -43% | 15% | -55% | -44% | -123% | -136% |
| Max RD | 86% | 123% | 147% | 157% | 125% | 35% | -27% |
| r[b] | 0.61 | 0.90 | 0.88 | 0.94 | 0.52 | 0.20 | 0.71 |
| **MFLEI versus GFED 4.1s (2003–2015)** | | | | | | | |
| Mean RD | 76% | 76% | 61% | 137% | 71% | 59% | 60% |
| Min RD | 50% | 58% | 29% | 104% | -24% | -29% | 18% |
| Max RD | 99% | 98% | 106% | 158% | 136% | 119% | 94% |
| r | 0.94 | 0.97 | 0.98 | 0.97 | 0.65 | 0.70 | 0.73 |
| **MFLEI versus WFEIS v0.5 (2003–2013)** | | | | | | | |
| Mean RD | 49% | 98% | 96% | 151% | 66% | 103% | 82% |
| Min RD | 19% | -59% | -154% | 63% | -118% | -174% | -86% |
| Max RD | 104% | 167% | 161% | 198% | 59% | 122% | 183% |
| r | 0.98 | 0.42 | -0.15 | 0.90 | 0.23 | -0.33 | 0.11 |

[a]

$$RD = 100 \times \frac{X(t)_{MFLEI} - Y(t)_i}{0.5 * (X(t)_{MFLEI} + Y(t)_i)}$$

X(t)$_{MFLEI}$ = MFLEI PM$_{2.5}$ emitted in year = t

Y(t)$_i$ = i PM$_{2.5}$ emitted in year = t, where i = FINN, GFED, or WFEIS

[b]r = correlation coefficient